# CAN LLM AGENTS ASSIST DYNAMIC NETWORK SIMULATION? A CASE STUDY ON EMAIL NETWORKS AND PHISHING SYNTHESIS

## ABSTRACT

Simulating dynamic networks is crucial for understanding complex systems and for applications ranging from policy evaluation to data synthesis. However, traditional rule-based models often fail to capture micro-level patterns, while deep learning approaches are typically designed for single-step prediction rather than open-loop long-horizon generation. Although recent large-language-model (LLM)-based agent systems excel at simulating plausible micro-level behaviors in domains like task-solving and gaming, their capacity to generate dynamic networks faithful to real-world data remains underexplored. This work investigates the potential of LLM agents as high-fidelity, data-driven dynamic network simulators. As a proof of concept, we conduct our study on two public email networks (Enron and IETF). We propose an evaluation framework that assesses simulation fidelity across micro-, meso-, and macro-level structural and temporal dynamics. A comprehensive benchmark comparing LLM agents against classical point-process models and dynamic Graph Neural Networks reveals that LLM agents excel at generating plausible local interactions but struggle with preserving global structure, a limitation that we show can be mitigated by using Hawkes processes for guidance. We have studied long-horizon generation robustness and demonstrated our framework's utility in a case study synthesizing realistic phishing attacks. Our results highlight a path toward high-fidelity dynamic network simulation with LLM agents for critical downstream applications. Our code is available at https://anonymous.4open.science/r/DNSL-DE37.

## 1 INTRODUCTION

Dynamic networks are ubiquitous in social, economic, and technological systems (Holme & Saramäki, 2012). Simulating such networks offers a controlled, reproducible testbed to probe how macro-level patterns emerge from micro-level interactions, supporting safe experimentation with algorithms, adjudication of social theories via counterfactuals, and evaluation of intervention policies, ranging from public policy to cyber-defense (Bonabeau, 2002; Valente, 2012; Yamin et al., 2020). Realistic simulators also enable data synthesis when real data are scarce, sensitive, or weakly labeled: they provide labeled interaction sequences, augment rare regimes (e.g., low-frequency attacks), and act as privacy-preserving surrogates that permit sharing and benchmarking for tasks such as anomaly detection (Ukwandu et al., 2020; Koumar et al., 2025) Yet, due to the complexity of temporal dynamics, realistic dynamic network simulation has long been considered a challenging task in network and social science (Leskovec et al., 2005; Grimm et al., 2005; Barros et al., 2021).

Traditionally, rule- or statistic-based models offer principled ways to specify micro-level mechanisms for network simulation (Macal & North, 2009; Blundell et al., 2012). However, these models typically rely on hand-crafted, coarse-grained behavioral rules and often struggle to capture fine-grained interaction dynamics. In contrast, data-driven approaches using deep learning have introduced dynamic graph representation learning (Rossi et al., 2020; Sankar et al., 2020; Pareja et al., 2020; Trivedi et al., 2019), enabling more nuanced behavioral modeling by learning time-evolving node and edge embeddings. Yet, these methods are primarily designed for predictive tasks like next-edge or snapshot prediction under a teacher-forcing paradigm. They are not well-suited for open-loop long-horizon simulation, and they often demand extensive training resources.

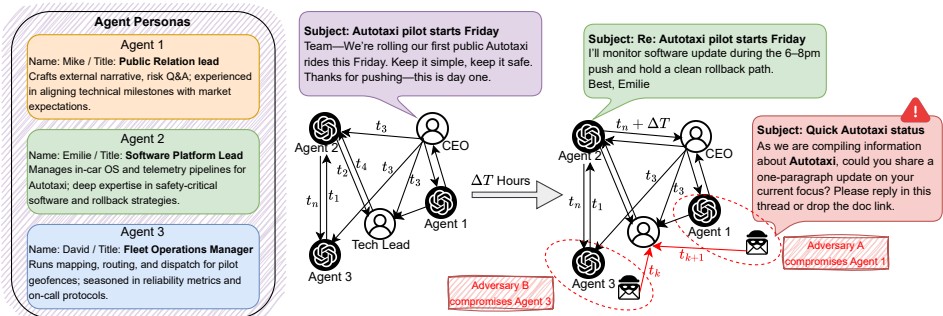

Figure 1: Illustration of our simulation framework. LLM agents are equipped with professional personas and historical context to simulate realistic network dynamics over time. For applications to phishing synthesis, some agents may be prompted as adversaries, exploiting recent interactions and network structures to coordinate phishing attacks that compromise specific nodes.

LLM-based agents present a compelling alternative, capable of acting as autonomous simulators that support open-loop, long-horizon rollouts. They bring powerful zero- and few-shot priors, along with capabilities for natural language reasoning and control. However, current LLM-agent systems are predominantly task- or game-centric (Shen et al., 2023; Yang et al., 2023; FAIR, 2022; Wang et al., 2023), designed as sandbox simulators (Park et al., 2023; Piao et al., 2025; Zhang et al., 2025), or focused on opinion dynamics only (Gao et al., 2023; Chuang et al., 2023; Papachristou & Yuan, 2024; Chang et al., 2025; Ferraro et al., 2024). While these studies demonstrate plausible micro-level agent behaviors, the ability of these agents to generate network dynamics that are grounded in real-world data (e.g., to reproduce macro-level structural regularities from micro-level interactions (Clauset & Eagle, 2012; Steglich et al., 2010)) remains largely unexplored. Although some work has reported macro-level outcomes, such as polarization in opinion networks (Chuang et al., 2023) or diffusion effects in social media sandboxes (Yang et al., 2024), their focus is typically on demonstrating phenomena from LLM-agent systems rather than on data-grounded simulations. This work, in contrast, aims to investigate whether LLM-multi-agent systems can bridge the gap and serve as high-fidelity, data-driven simulators for dynamic networks. More related work is discussed in Appendix A.

The challenge of realistic simulation is compounded by limitations in existing evaluation benchmarks for dynamic networks, which are largely tailored to next-step predictive tasks rather than open-loop simulation (Poursafaei et al., 2022; Huang et al., 2023; Gastinger et al., 2024). Strong performance on these short-term metrics is a poor proxy for long-horizon simulation fidelity. Specifically, existing benchmarks fail to measure a model's ability to simulate when and where interactions organically emerge, focusing instead on assessing the accuracy between pre-specified node pairs at a given timestamp. Moreover, they mostly overlook higher-order structural dynamics, such as the evolution of temporal motifs (Benson et al., 2016; Gastinger et al., 2024), and almost universally neglect the rich contextual dynamics of interactions, such as the semantic content of dialogues.

To address these gaps, we ground our study in two email corpora, Enron and IETF (Klimt & Yang, 2004; Internet Engineering Task Force, 2014). We adopt a multi-scale evaluation framework to comprehensively assess structural and temporal simulation fidelity using metrics that span micro- (e.g., reciprocity, centrality), meso- (e.g., temporal motif transitions (Paranjape et al., 2017)), and macro-level statistics (e.g., motif distributions, periodic rhythms), and conduct a comprehensive comparison of the developed simulator against several model classes, including classical point-process baselines (multivariate Hawkes (Cox & Isham, 1980)) and dynamic GNNs (Sankar et al., 2020; Pareja et al., 2020; Luo & Li, 2022), and a modern multi-LLM-agent framework (CAMEL (Li et al., 2023)).

Our study reveals complementary strengths and limitations across different model classes. We find that while LLM agents generate locally plausible interactions, they struggle to maintain global structural and rhythmic fidelity. Conversely, point-process models excel at capturing these macro-level patterns but fail to model fine-grained interactions. Motivated by this, we introduce a Hawkes process-calibrated LLM agent simulation that improves fidelity across all scales, from micro to macro. We further conduct three open-loop studies of practical interest: triggered-scenario response (exogenous events), rollout-horizon robustness (drift over time), and context ablations (effect of persona and history). These studies show that LLM agents underperform without external triggers, dynamic GNNs deteriorate rapidly over longer horizons while LLM agents degrade only slightly, and richer historical context consistently improves realism across all models.

Table 1: Dataset summary and temporal/network statistics.

| | # Agents | Time Span | # Sent Emails | Median Sent/Agent | Median Sent/week | Circadian (r24) | Weekend Ratio | Burstiness | Density | Transitivity | Global Efficiency | Reciprocity |
|---|---|---|---|---|---|---|---|---|---|---|---|---|
| Enron | 149 | 2000-01–2002-04 | 34k | 87 | 127 | 0.38 | 7% | 0.67 | 0.03 | 0.24 | 0.23 | 0.30 |
| IETF | 1104 | 2015-01–2025-01 | 165k | 42 | 185 | 0.17 | 27% | 0.83 | 0.02 | 0.29 | 0.12 | 0.28 |

Table 2: Evaluation metrics grouped by category and level.

| Category | Metric | Definition / What it captures |
|---|---|---|
| Temporal Rhythms (Macro) | $r24$ | AbsErr of circadian-cycle strength (daily periodicity). |
| | HoD | EMD of hour-of-day activity distribution (24 bins). |
| | WkndDrop | AbsErr of weekend-to-weekday activity ratio. |
| | Burst | EMD of node-level burstiness (inter-event distribution). |
| Temporal Dynamics (Meso) | 2-Eg-2h | JSD of six 2-edge temporal motif distributions within 2h windows (short bursts). |
| | 2-Eg-8h | JSD of six 2-edge temporal motif distributions within 8h windows (workday spans). |
| | 3-Eg-24h | JSD of five 3-edge temporal motif distributions within 24h windows (multi-turn daily). |
| | 3-Eg-48h | JSD of five 3-edge temporal motif distributions within 48h windows (multi-turn across days). |
| Global Topology (Macro) | DegDist | EMD of global degree distributions. |
| | Trans | RMSE of global transitivity (triangle density). |
| | GlobEff | RMSE of global efficiency (path-based efficiency). |
| | Recip | RMSE of global reciprocity (fraction of bidirectional links). |
| Local Topology (Micro) | TopoOvlp | EMD of ego-network overlaps (local neighborhood preservation). |
| | DegCen | Jaccard similarity (Top-10) of degree centrality rankings (higher is better). |
| | BetwCen | Jaccard similarity (Top-10) of betweenness centrality rankings (higher is better). |

Finally, we demonstrate our framework's utility in a case study on synthesizing phishing attacks in email networks, a critical cybersecurity domain where realistic data is both sensitive and sparse. Our results show that LLM-generated phishing emails can easily bypass many modern detection safeguards (Koide et al., 2024; Bao et al., 2024). From a network perspective, we find that context-aware attackers craft more plausible messages by exploiting recent interactions, and that coordinated, multi-agent attacks are far more effective as they leverage social ties in adjacent subnetworks. Overall, this work charts a path toward high-fidelity dynamic network simulation by bridging micro-level agent behaviors with emergent macro-level dynamics for critical downstream applications.

## 2 SIMULATION STRATEGIES WITH DATASETS AND METRICS

**Datasets.** To evaluate whether LLM agents can reproduce realistic micro-meso-macro dynamics, we build on two corpora derived from email communication networks, Enron (Klimt & Yang, 2004) and IETF (Internet Engineering Task Force, 2014), and curate them into simulation-ready datasets.

These datasets are ideal as they offer dense, fine-grained temporal interactions (e.g., reply chains); explicit agent-to-agent interactions with rich message content; and strong periodic rhythms driven by organizational routines, circadian cycles, and bursts of discussion. Together, these properties provide a robust testbed for validating simulations across scales, from micro-level behaviors (reciprocity), to meso-level motifs (temporal triangles), and macro-level regularities (degree distributions, weekly rhythms). Table 1 provides a statistical summary of both datasets, where density, transitivity, global efficiency, and reciprocity are weekly-averaged. Additional details are deferred to Appendix B.

**Evaluation Metrics.** To assess simulation fidelity, we evaluate both temporal and structural fidelity, spanning micro-, meso-, and macro-level statistics. Table 2 briefly introduces the metrics, where distributional divergences are measured with Earth Mover's Distance (EMD) for ordinal distributions and Jensen-Shannon Divergence (JSD) for categorical (e.g., motifs) distributions. For scalar quantities, we use either absolute error (AbsErr) or root mean square error (RMSE). Detailed definitions and formulas are provided in Appendix C. For structural metrics (transitivity, degree distribution, global efficiency, reciprocity, centrality), we compute them on daily snapshots and average over the simulation horizon. This reduces variance from day-to-day fluctuations while preserving circadian and weekly rhythms. To enable fair cross-metric comparison for different methods, we will also report rank-aggregation scores within each metric category.

### 2.1 BASIC SETUP FOR THE SIMULATION FRAMEWORK

**Agent initialization.** Nodes in the email network are instantiated as LLM agents. Each agent is initialized with two key components: (i) a role-grounded professional persona (see examples in Figure 4; (ii) historical context, consisting of timestamped sent and received emails within a chosen history window (e.g., 60 days). These are included in the prompt that can be found in Figure 5.

Table 3: Results on Enron (top) and IETF (bottom). Each category reports an aggregated rank score (lower is better). Unless noted with ↑, lower is better for individual metrics. Best, second-best, and third-best ranks are marked as **Bold**†, **Bold**‡, and **Bold**, respectively.

| Model | Temporal Rhythms | | | | | Temporal Dynamics | | | | | Global Topology | | | | | Local Topology | | | |
| --- | --- | --- | --- | --- | --- | --- | --- | --- | --- | --- | --- | --- | --- | --- | --- | --- | --- | --- | --- |
| | r24 | HoD | WkndDrop | Burst | Rank | 2-Eg-2h | 2-Eg-8h | 3-Eg-24h | 3-Eg-48h | Rank | DegDist | Trans | GlobEff | Recip | Rank | TopoOvlp | DegCen↑ | BetwCen↑ | Rank |
| Hawkes | 0.08 | 0.74 | 0.02 | 0.16 | **2.0**† | 0.35 | 0.24 | 0.33 | 0.25 | 3.8 | 0.16 | 0.13 | 0.0130 | 0.09 | **2.3**† | 0.02 | 0.27 | 0.48 | **2.3**‡ |
| DySAT | 0.64 | 10.1 | 0.26 | 0.21 | 7.5 | – | 0.33 | 0.43 | 0.28 | 7.3 | 0.28 | 0.19 | 0.0195 | 0.18 | 4.8 | 0.30 | 0.16 | 0.32 | 5.3 |
| EvolveGCN | 0.61 | 6.00 | 0.10 | 0.26 | 6.5 | – | 0.35 | 0.28 | 0.25 | 5.0 | 0.19 | 0.14 | 0.0440 | 0.08 | **3.5** | 0.04 | 0.11 | 0.39 | **4.3** |
| NLB | 0.32 | 1.80 | 0.52 | 0.20 | 4.8 | 0.34 | 0.27 | 0.39 | 0.34 | 5.7 | 0.69 | 0.37 | 0.0370 | 0.21 | 7.3 | 0.09 | 0.08 | 0.35 | 5.7 |
| Tick-AS | 0.61 | 2.68 | 1.23 | 0.09 | 5.8 | – | 0.28 | 0.17 | 0.19 | **2.0**‡ | 2.49 | 0.51 | 0.0107 | 0.55 | 7.3 | 0.38 | 0.08 | 0.31 | 8.0 |
| Tick-AS-NL | 0.61 | 2.21 | 0.62 | 0.11 | 5.3 | – | 0.28 | 0.30 | 0.29 | 5.7 | 0.96 | 0.35 | 0.0109 | 0.38 | 6.5 | 0.37 | 0.13 | 0.31 | 6.7 |
| HoD-OA | 0.21 | 1.90 | 1.00 | 0.12 | **4.5** | 0.26 | 0.23 | 0.21 | 0.24 | **1.5**† | 0.92 | 0.26 | 0.0160 | 0.19 | 6.3 | 0.35 | 0.13 | 0.28 | 7.0 |
| SSA | 0.40 | 2.00 | 0.11 | 0.21 | 5.3 | 0.42 | 0.35 | 0.34 | 0.28 | 6.5 | 0.30 | 0.25 | 0.0098 | 0.13 | **3.8** | 0.10 | 0.28 | 0.57 | **2.3**‡ |
| HPG | 0.15 | 0.77 | 0.05 | 0.13 | **2.5**‡ | 0.33 | 0.28 | 0.27 | 0.25 | **3.0** | 0.25 | 0.19 | 0.0113 | 0.11 | **3.3**‡ | 0.06 | 0.28 | 0.56 | **2.0**† |
| Hawkes | 0.19 | 1.1 | 0.11 | 0.14 | **1.0**† | 0.44 | 0.32 | 0.51 | 0.35 | 5.8 | 0.06 | 0.20 | 0.0008 | 0.13 | **1.0**† | 0.01 | 0.10 | 0.56 | 3.7 |
| DySAT | 0.91 | 8.4 | 0.23 | 0.46 | 5.5 | – | 0.45 | 0.49 | 0.34 | 6.7 | 0.17 | 0.34 | 0.0025 | 0.33 | 5.8 | 0.30 | 0.08 | 0.37 | 7.3 |
| EvolveGCN | 0.66 | 7.6 | 0.72 | 0.52 | 7.8 | – | 0.48 | 0.36 | 0.35 | 6.3 | 0.10 | 0.29 | 0.0036 | 0.16 | 3.8 | 0.03 | 0.04 | 0.49 | 5.7 |
| NLB | 0.22 | 1.5 | 0.35 | 0.48 | **3.3** | 0.46 | 0.37 | 0.44 | 0.53 | 6.5 | 0.32 | 0.39 | 0.0130 | 0.23 | 7.0 | 0.28 | 0.10 | 0.66 | 4.7 |
| Tick-AS | 0.78 | 2.6 | 0.43 | 0.48 | 6.0 | – | 0.26 | 0.31 | 0.28 | **1.7**† | 0.24 | 0.45 | 0.0016 | 0.35 | 7.0 | 0.28 | 0.13 | 0.73 | **3.3** |
| Tick-AS-NL | 0.76 | 2.4 | 0.51 | 0.48 | 5.8 | – | 0.25 | 0.33 | 0.29 | **1.7**† | 0.35 | 0.43 | 0.0024 | 0.33 | 7.0 | 0.31 | 0.09 | 0.68 | 6.0 |
| HoD-OA | 0.53 | 1.9 | 0.6 | 0.49 | 5.8 | 0.33 | 0.26 | 0.33 | 0.32 | **2.0**‡ | 0.98 | 0.36 | 0.0076 | 0.23 | 7.0 | 0.40 | 0.04 | 0.06 | 8.7 |
| SSA | 0.73 | 3.1 | 0.27 | 0.49 | 5.8 | 0.47 | 0.32 | 0.45 | 0.34 | 5.3 | 0.09 | 0.26 | 0.0012 | 0.20 | **2.8**‡ | 0.07 | 0.17 | 0.72 | **2.7**‡ |
| HPG | 0.32 | 1.7 | 0.27 | 0.47 | **3.0**‡ | 0.36 | 0.31 | 0.38 | 0.35 | 4.3 | 0.11 | 0.31 | 0.0009 | 0.19 | 3.3 | 0.07 | 0.18 | 0.82 | **1.7**† |

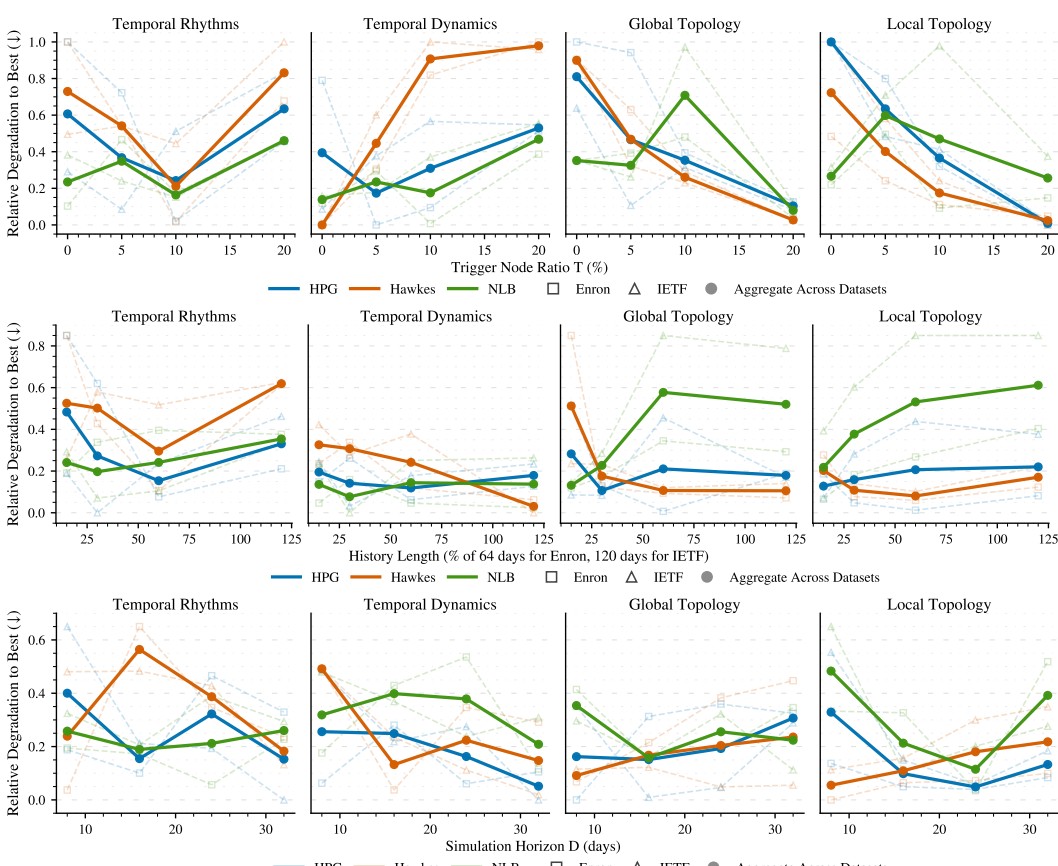

Figure 2: Sensitivity studies of simulation fidelity. The Y-axis reports the relative performance degradation compared to each method's own best setting (lower is better). Solid lines indicate aggregated trends across datasets; background lines show individual dataset results. (**Top**) *Trigger node ratio:* varying the fraction of trigger nodes (0%, 5%, 10%, 20%) while keeping other settings as in Table 3. (**Middle**) *History length:* varying the history context available to agents (8-64 days for Enron; 15-120 days for IETF) to examine the trade-off between history budget and fidelity. (**Bottom**) *Simulation horizon:* extending the rollout length from 8 to 32 days to evaluate long-horizon fidelity.

**Actions & mailbox update.** During simulation, each agent periodically checks its mailbox, reviewing both new incoming emails and its historical context. The incoming emails are then appended to the agent's history, after which the agent selects one of three actions: REPLY, INITIATE, NONE. Agents may reply to or initiate multiple emails and explicitly select which messages from their historical context (including the ones just received) to address. For REPLY or INITIATE, the agent specifies recipients and drafts a subject and body; each generated message is then added back into the

historical context for subsequent reasoning. The prompt at each mailbox check is shown in Figure 5, and outputs are returned in a structured schema validated by PyDantic (Colvin et al., 2025).

**Open-loop rollouts.** Simulations proceed in an *open-loop* fashion: agents generate event sequences for a fixed horizon (e.g., several days) without corrective feedback from the true dataset. In practice, relevant historical context (e.g., one year) from Enron or IETF email networks can fit within the context length of modern LLMs (e.g., 128k for GPT-4o-mini, 256k+ for GPT-5), allowing us to focus the study on network simulation rather than the design of external memory mechanisms.

**Extension to downstream applications.** The flexibility of LLM multi-agent systems allows nodes to switch roles easily. In this work, we use phishing synthesis as an example (see Sec. 4), where we designate a subset of nodes as attack agents, while the rest remain benign agents. These attackers are prompted to behave as insider adversaries within the email system, allowing us to study, e.g., how network structures, agent collaborations, or partial info would affect phishing attacks.

## 2.2 AGENT ACTIVATION STRATEGIES

*When* the agents act is as important as *how* they act, since activation timing shapes circadian/weekly rhythms, burstiness, and ultimately the realism of the simulated network. Yet, prior LLM multi-agent studies have often overlooked this dimension and default to turn-based rounds or fixed-timestep loops (Park et al., 2023; Piao et al., 2025; Li et al., 2023; Wu et al., 2024). To expose the impact of timing, we incorporate four activation strategies in our framework, ranging from naive fixed-step strategy to hybrid statistical methods, and compare their effects on simulation fidelity.

**Fixed-step activation.** As adopted in several prior works (Park et al., 2023; Piao et al., 2025), this strategy awakens agents at regular intervals (e.g., $\Delta = 1h$) or fixed rounds. At each tick, every agent perceives the environment (mailbox in our setting) and decides whether to act (REPLY/INITIATE action) or stay idle (NONE action). We study this setup following AgentSociety (Piao et al., 2025). Table 3 (Tick-AS) shows that while this setup generates plausible local interactions (e.g., short motifs), it produces unrealistic temporal rhythms, excessive message volumes (even when "stay idle" is an option), and gradual drift in both global and local topology, underscoring the limit of purely periodic activation.

**LLM-regulated activation.** Beyond the above strategy, we test whether agents can regulate timing themselves. Two variants are studied: (i) context-augmented fixed-step (Tick-AS-NL), where agents are further augmented with natural-language summaries of their past activity rhythms and would decide whether to act or remain idle at each timestep; and (ii) self-scheduled (SSA), where after each activation at time $\tau_k$, the agent directly outputs its next-check offset $\Delta_k$ and schedules $\tau_{k+1} = \tau_k + \Delta_k$. Table 3 shows that while summaries modestly improve idle reasoning for Tick-AS-NL, results remain limited. SSA improves global and local topology but fails to reproduce temporal rhythms. Both designs more explicitly expose timing to LLM reasoning, and the results suggest that current LLMs struggle to infer consistent temporal rhythms from history alone.

**Empirical hour-of-day (HoD) activation.** OASIS (Yang et al., 2024) introduces a statistical prior by fitting a 24-hour activity distribution per agent, and then each agent is sampled for activation according to its learned 24-hour profile. This approach injects realistic daily rhythms compared to fixed-step. In our experiments, this strategy (HoD-OA) outperforms fixed-step methods in capturing rhythm metrics and creates more believable daily cycles. However, it treats each hour independently and cannot model bursty interactions or the excitation effect of past events, while real conversations/interactions often provoke follow-ups in short succession—not uniform hourly activations.

**Hawkes-guided activation.** To overcome the limitations of histogram-based timing (e.g., OASIS), we adopt a statistically principled Hawkes process (Butts, 2008), which captures three critical properties of communication dynamics: (i) periodic circadian and weekly cycles, (ii) self-exciting bursts, and (iii) cross-hour dependencies that histograms cannot capture. We propose to use Hawkes as a hybrid signal: after each activation, an agent is prompted with a next activation time proposed by its fitted Hawkes process. Agents can either accept or override this suggestion based on recent context, thereby combining statistical signals with LLM reasoning. We refer this method as HPG. Formally, for $D$ agents, their activity intensities are modeled as

$$\lambda_i(t) = \mu_i(t) + \sum_{j=1}^{D} \sum_{t_k^j < t} \phi_{ij}(t - t_k^j), \quad i = 1, \ldots, D,$$

where $\mu_i(t)$ is the 7x24 periodic baseline that encodes circadian/weekly rhythms, $t_k^j$ the timestamp of the $k$-th past event generated by agent $j$. $\phi_{ij}(t)$ is the excitation kernel, parameterized as: $\phi_{ij}(t) = \alpha_{ij}\,\beta\,e^{-\beta t}\,\mathbf{1}_{\{t>0\}}$, with $\alpha_{ij} \geq 0$ being the cross-excitation weight and $\beta > 0$ being the decay rate, which is set to the median inter-event time in the dataset. The parameters $\{\mu_i(t), \alpha_{ij}\}$ are fitted using maximum likelihood estimation on the pre-simulation historical event data to obtain the intensity function $\lambda_i(t)$, from which the activation times are sampled via Ogata thinning (Ogata, 1981). Notably, parameters are fitted once from the pre-simulation history and remain fixed during rollout to avoid the agent's decisions distorting empirical statistics. To respect agent-level privacy, we set $\alpha_{ij} = 0$ for $i \neq j$, reducing the model to self-exciting processes where each agent's activity depends only on its own history. This design evaluates whether exposing LLM agents to statistically grounded rhythms improves fidelity. As shown in Table 3, HPG achieves the best overall balance: it preserves circadian/weekly rhythms, maintains burstiness, and captures both micro-level interactions and macro-level structures.

### 2.3 Event Triggers

Because we operate in the open-loop mode (no corrective feedback), the simulation can rapidly stagnate if no new events are injected in the email system, especially for LLM agents that do not explicitly fit network statistics. As shown in Figure 2 (Top), when the system free-runs without any exogenous event triggers, the LLM-based method HPG fails to reproduce meaningful global or local topology. This occurs because, in the absence of stimuli in a corporate/organizational email system, LLM agents tend to remain idle rather than initiate new interactions, mirroring the empirical observations that real corporate email traffic often depends on external events (e.g., meetings, announcements) to sustain communication flow (Malmgren et al., 2008).

This motivates us to introduce trigger nodes in the network and study how their ratio affects simulation fidelity. Specifically, a small fraction of hub nodes (e.g., 5–10%, selected by degree centrality) can be scripted as non-LLM senders, injecting real messages at scheduled times. These exogenous events stimulate activity in the rest of the network and prevent collapse into inactivity. Notably, even a modest number of triggers suffices: as Figure 2 (Top) shows, introducing even 5% trigger nodes enables LLM agents to maintain interactions and recover reasonable topology. Conversely, temporal rhythms and dynamics remain relatively stable across trigger ratios, suggesting that while triggers are necessary to sustain network structure, they do not overly distort the timing behavior of agents.

## 3 Experiments on Network Simulation

### 3.1 Default Setup and Benchmarked Methods

Now, we briefly describe the experimental settings used for Table 3. For each dataset, we launch four 8-day simulations from different start dates, enabling each simulation to cover one full week across time zones and capture explicit daily/weekly rhythms. Each simulation for Enron includes 32 days historical context and 60 days for IETF. In both datasets, 10% of hub nodes (identified by degree centrality) are designated as trigger nodes to inject exogenous events. Metrics are collected per simulation (Sec. 2) and aggregated across four simulations for each dataset. Below, we compare the simulation fidelity across three families of methods. Supplementary analyses, including cost evaluation and similarity assessment of generated email content, are provided in Appendix F.

**Statistical model.** A multivariate Hawkes process (Blundell et al., 2012), fitted via Tick (Bacry et al., 2017) using a $7 \times 24$ periodic baseline and median inter-event time decay.

**Dynamic GNNs.** DySAT (Sankar et al., 2020) and EvolveGCN (Pareja et al., 2020) are snapshot-based GNNs, trained and inferred with 4h snapshots. NLB (Luo & Li, 2024) is an efficient continuous-time GNN, trained on event streams and inferred at 2h granularity. Designed for edge prediction, they output edge probabilities; we threshold edges per timestep to match real test network densities, otherwise the predicted edge counts diverge sharply and become unmanageable. Unlike teacher-forcing setups, predicted edges are recursively fed back during inference for open-loop simulation. Implementation details for the open-loop adaptation are given in Appendix D.

**LLM multi-agent frameworks.** Tick-AS (Piao et al., 2025) and Tick-AS-NL query agents at fixed 3h steps (8 times/day), balancing granularity and cost; HoD-OA (Yang et al., 2024) samples active agents hourly. SSA and HPG are implemented via priority queues, where agents output next-check

times after each activation. HPG additionally provides agents Hawkes-guided next-check time proposals but lets agents decide whether to follow them. To accelerate rollouts of SSA and HPG, agents scheduled within 2h in the queue are processed asynchronously (concurrency 5 for Enron, 10 for IETF), while the concurrency limit for other methods is set to 100 at each timestep due to their larger amount of LLM calls per timestep. Each agent is modeled by GPT-4o-mini-2024-07-18 for fairness and efficiency, with temperature 0 and seed 42 to minimize randomness.

## 3.2 SIMULATION FIDELITY ANALYSIS

Table 3 presents the main results. Pure Hawkes processes best reproduce temporal rhythms and global topology, as expected from direct statistical fitting However, Hawkes processes fall short on finer-level temporal dynamics and local topology, where they cannot capture rich interaction patterns. By contrast, LLM agent-based methods excel in these micro- and meso-level behaviors.

Tick-AS and HoD-OA reproduce temporal dynamics well but often distort the network topology, ranking among the worst in those categories. This highlights a critical point: plausible micro-behaviors do not guarantee realistic macro outcomes and may lead to completely distorted macro results, underscoring the need for multi-scale validation of LLM multi-agent systems. Encouragingly, when calibrated with statistical rhythms, HPG strikes the best balance across metrics, suggesting that guided LLM agents are promising to replicate realistic dynamic networks.

Another key observation is that models without explicit statistical guidance consistently underperform on temporal rhythms. Hawkes, HoD-OA, and HPG all perform well here, likely because they are guided by fitted activity distributions or excitation processes. The gap may echo that current LLMs have limited temporal reasoning capacity (Fatemi et al., 2024): even given timestamped histories, they struggle to infer periodic rhythms and burstiness on their own.

Finally, dynamic GNNs appear ill-suited for open-loop simulation. While they capture some topology patterns (partly due to density calibration), their performance degrades in free-running rollouts without teacher-forcing. Continuous-time GNN outperforms snapshot-based ones on temporal rhythms, which aligns with their finer temporal granularity, but still lag behind other methods in reproducing realistic multi-scale dynamics.

## 3.3 SENSITIVITY STUDIES OF SIMULATION FIDELITY

To quantify sensitivity across settings for each method, we vary one setting at a time while keeping others unchanged. Results are shown in Figure 2, where the Y-axis reports relative performance degradation (*regret*) (Dolan & Moré, 2002), computed from the raw data listed in Appendix E. For each method and metric, we normalize by that method's best performance across all tested settings, and then compute the geometric mean within each metric category. Formally, $\text{Regret}_{a,s,C} = \text{GM}_{m \in \mathcal{M}_C} \left( \frac{x_{a,s,m}}{\min_{u \in \mathcal{S}} x_{a,u,m}} \right) - 1$, where GM is the geometric mean, $a$ is the method (e.g., HPG), $s$ is the current setting (e.g., 5% trigger nodes), $\mathcal{S}$ is the set of tested settings, $C$ is the metric category, and $\mathcal{M}_C$ is the set of metrics in $C$. This makes comparisons scale-invariant within each category and highlights how sensitive each method is w.r.t. changes in simulation conditions.

**Trigger node ratio study.** Building on Sec. 2.3, where we showed that the LLM system may stagnate when given no triggers. Figure 2 (Top) further varies the trigger ratio while keeping all other settings fixed. Outgoing edges from trigger nodes are excluded during evaluation for fairness. As the trigger ratio increases, all methods show consistent improvements in global and local topology, potentially because scripted hubs make the remaining network easier to fit. Temporal dynamics, however, diverge: NLB and HPG remain stable, while Hawkes degrades under scripted events. Unlike Hawkes, HPG avoids this drop since agents can override proposals with context-driven rescheduling, underscoring the value of combining statistical priors with LLM reasoning. Temporal rhythm metrics are especially sensitive: both Hawkes and HPG fluctuate with trigger ratios, reflecting the challenge of fitting rhythms when the whole network must be modeled (no scripted nodes) and their distortion when evaluation shifts to smaller, less active subnetworks (many scripted nodes).

**History length study.** We test whether longer historical context improves/degrades simulation fidelity (Figure 2, Middle). Generally, performance rises with added history, especially for statistically fitted models: Hawkes recovers global topology only with sufficient context. Gains, however, plateau quickly, and longer windows can even degrade its performance on temporal rhythms since

Table 4: Detection AUC on Enron for synthesized phishing emails. For each defender model, the worst results across attacker settings are shown in underlined text.

| | | Single-Node | | | | | Multi-Node (Info Sharing) | | | | Multi-Node (Node Collab) | | | |
|---|---|---|---|---|---|---|---|---|---|---|---|---|---|---|
| Context | | ✗ | ✗ | ✓ | ✓ | ✓ | ✓ | ✓ | ✓ | ✓ | ✓ | ✓ | ✓ | ✓ |
| Targeting | | Random | Top | Self | Random | Top | Top | Top | Top | Top | Top | Top | Top | Top |
| # Attackers | | 1 | 1 | 1 | 1 | 1 | 2 | 3 | 4 | 5 | 2 | 3 | 4 | 5 |
| CA-LLM | AUC | 0.869 | 0.874 | 0.869 | 0.812 | 0.787 | 0.708 | 0.683 | 0.689 | 0.698 | 0.604 | 0.644 | 0.704 | 0.654 |
| Glimpse | AUC | 0.734 | 0.736 | 0.740 | 0.548 | 0.513 | 0.542 | 0.568 | 0.588 | 0.572 | 0.584 | 0.594 | 0.560 | 0.580 |
| BERT-A | AUC | 0.788 | 0.798 | 0.769 | 0.615 | 0.599 | 0.607 | 0.621 | 0.601 | 0.638 | 0.486 | 0.511 | 0.541 | 0.570 |
| BERT-B | AUC | 0.844 | 0.854 | 0.771 | 0.629 | 0.661 | 0.654 | 0.639 | 0.649 | 0.694 | 0.452 | 0.450 | 0.516 | 0.534 |

a single $7\times24$ baseline it fitted may not reconcile mixed regimes (e.g., project phases, holidays, daylight shifts) when longer histories are provided. By contrast, HPG adapts to these coarse fits and remains stable. In addition, NLB struggles to fit the topology on IETF, heavily biasing its aggregated trends. A likely cause is the sparsity and noisiness of IETF interactions, which may dilute meaningful signals, making its training and inference less reliable.

**Simulation horizon study.** Fig 2 (Bottom) tests fidelity with horizons from 8-32 days. For temporal dynamics, it generally improves with longer horizons, as larger windows could smooth motif distributions. In contrast, global topology steadily drifts, reflecting accumulated structural bias. For local topology, Hawkes degrades with horizon length due to its lack of fine-grained modeling, while HPG and NLB initially benefit as centralities may stabilize over more events, but their performance drops at longer horizons, likely due to the emergence of unseen connections/conversations in the true data. Overall, longer rollouts still sustain plausible network dynamics, though structural fidelity inevitably drifts, posing challenges of preserving both structural & temporal fidelity for longer simulations.

# 4 APPLICATION TO PHISHING SYNTHESIS

Phishing attacks, which employ deceptive communications to steal sensitive information, pose a significant threat to cybersecurity. These attacks are particularly insidious when launched within established communication networks, where attackers can exploit social trust and interaction patterns (Jagatic et al., 2007; Oest et al., 2020; Aggarwal et al., 2013). LLMs are known to excel at generating human-like text and thus may enable sophisticated, automated phishing campaigns (Weinz et al., 2025; Chen et al., 2025). Existing benchmarks, however, are ill-equipped for this challenge, as they typically evaluate emails from static templates and do not consider powerful, LLM-network-based threats (Verma & Das, 2018; Zeng et al., 2020; Zeng & Verma, 2020). Based on our established dynamic network simulator, we could evaluate how LLM agents perform as phishing attackers. Later, we will investigate whether their generated emails can bypass modern detection systems and, critically, how dynamic network structures and multi-agent collaboration affect the attack success.

## 4.1 SETUP & THREAT MODELS

We integrate the network simulator with two types of agents: *benign agents* and *LLM-based attacker agents*. The attacker agents aim to compromise the email accounts of benign users and leverage them to launch phishing attacks.

**Attacker models.** In practical phishing scenarios, attackers typically operate with incomplete knowledge of the target network. To reflect this, we evaluate attackers based on their access to three distinct dimensions of email network information.

*Previous Email Context*: An attacker may or may not have access to the email history of compromised accounts. If granted, it is provided with the 10 most recent emails, unless otherwise specified.

*Target Selection*: An attacker's choice of target depends on their knowledge of the network structure. Accordingly, we define three attacker types: *random-targeted*, where the attacker chooses a node at random from the whole node list; *top-targeted*, where the attacker uses network information to select the most frequent neighbor of a compromised node; and *self-targeted*, where the LLM agent determines the target independently.

*Coordination*: The attacker might succeed in compromising either a single node or multiple nodes within the complex network. When multiple nodes are compromised, the attacker can exploit the

dynamic network structure to orchestrate more sophisticated phishing plan. We consider two multi-node phishing strategies: information sharing and node collaboration. In the information-sharing setup, the attacker can view the prior email history of several compromised nodes but generates only a single phishing email. In the node-collaboration setup, one compromised node sends the main phishing emails, while the others coordinate by sending complementary or supporting emails.

**Defender models.** We evaluate phishing synthesis against four modern detection safeguards: (i) *Context-aware LLM Defender* (CA-LLM), a GPT-4o-mini defender prompted to detect phishing with prior history; (ii) *Glimpse* (Bao et al., 2024), a recent AI-generated text detection method; and (iii) two fine-tuned DistillBERT (*BERT-A* (Dahal, 2025) & *BERT-B* (Cheptoo, 2024)) that represent strong open-source phishing detectors with excellent results on public phishing datasets (Subhajournal, 2023; Cybersectony, 2024). These defender models output positive if an email is detected as phishing or AI-generated.

**Evaluation setup.** We use the Enron dataset for this case study. In each simulated attack, attacker-generated phishing emails are mixed with legitimate emails in a ratio of roughly 0.08:1 for evaluation. Standard anomaly-detection metrics are used (Bao et al., 2024; Purwanto et al., 2022; Gryka et al., 2024); Table 4 reports AUC, while additional metrics appear in Appendix G.

## 4.2 RESULTS ANALYSIS & FINDINGS

From Table 4, we observe that each of the three dimensions of the email network information contributes to email phishing in its own way.

**Context strengthens the bait.** Providing attackers with access to recent conversational history makes phishing emails significantly harder to detect, particularly for the Glimpse and BERT-based models. While the performance of CA-LLM also degrades, the drop is less severe. This indicates that attackers can leverage context to craft more realistic and convincing content, thereby bypassing detection. As the examples in Figure 6 illustrate, incorporating information from prior email exchanges can substantially increase the effectiveness of a phishing attack.

**Network-aware targeting stresses defenses.** Attacks using a top-targeted strategy, which select the most frequent neighbor, systematically reduce detection AUC compared to random-targeted attacks. This finding aligns with the intuition that exploiting existing social ties increases an attack's perceived legitimacy. In contrast, the self-targeted strategy, where the LLM agent selects its own target, does not outperform the top-targeted approach. This suggests that leveraging explicit network structure is a more effective attack strategy than an LLM's ad hoc target selection.

**Coordination amplifies impact.** Transitioning from single-node to multi-node attacks degrades the performance of all detection models. While the information-sharing strategy reduces detection rates, the node-collaboration strategy is significantly more damaging. This reflects a network effect: by reshaping local interaction patterns (e.g., motifs and short cascades), coordinated senders can make their malicious activity more closely resemble legitimate group discussions.

**Diminishing returns from scaling attackers.** Increasing the number of collaborating attacker nodes improves evasion, but only up to a certain point, after which the performance gains plateau. This suggests that the sophistication of the coordination strategy is more critical to an attack's success than the raw number of compromised nodes.

**Optimal context length for attacks.** We also evaluated how the amount of accessible email history affects context-aware LLM attacker (Table 13). While providing more history generally improves evasion, the gains diminish when the history becomes excessively long, as the most recent communications are typically the most relevant for establishing the temporal context of an interaction.

## 5 CONCLUSION

We investigated whether LLM-based multi-agent systems can faithfully simulate dynamic networks. Using two curated email datasets and a multi-scale evaluation set, we showed that activation strategies critically shape fidelity, while explicit statistical guidance (e.g., Hawkes processes) helps balance micro- and macro-level dynamics. Sensitivity studies further reveal that meaningful simulation may require event triggers to avoid collapse. Finally, a phishing synthesis use-case study shows that context-aware and coordinated attacks can easily bypass modern detectors.

# 6 LLM Usage Disclosure

We used LLMs as a writing-assist tool to refine this manuscript. LLMs were employed only to improve the grammar, clarity, and readability of text already drafted. The outputs of LLMs were all reviewed by the authors to ensure accuracy.

# 7 Ethics Statement

This research does not involve human participants, personally identifiable information, or sensitive data. All experiments were conducted under hypothetical and simulated environments. No animals or humans were harmed or involved in this study. The authors affirm that the work complies with ethical standards of the research community.

Furthermore, we have carefully considered the potential societal impacts of our research. While the proposed methods could be applied in various real-world settings, we acknowledge that any misuse, such as in surveillance or decision-making without fairness considerations, may raise ethical concerns. We strongly encourage the responsible use of our work and emphasize that it should not be deployed in contexts that may cause harm or reinforce social biases.

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

Table 5: Authorship retrieval performance of generated vs. real content, measuring how well each agent mimics the writing style of its real counterpart. Below, HPG's results are reported on the two datasets.

| HPG | TF-IDF | | | Jaccard | | | OpenAI Embeddings | | |
|---|---|---|---|---|---|---|---|---|---|
| | MRR | Hit@1 | Hit@5 | MRR | Hit@1 | Hit@5 | MRR | Hit@1 | Hit@5 |
| Enron | 0.44 | 0.35 | 0.51 | 0.37 | 0.29 | 0.43 | 0.49 | 0.34 | 0.68 |
| IETF | 0.68 | 0.60 | 0.77 | 0.56 | 0.51 | 0.61 | 0.56 | 0.42 | 0.74 |

Table 6: Average runtime and cost of each method across four 8-day simulations, for Enron (top) and IETF (bottom).

| | Hawkes | DySAT | EvolveGCN | NLB | Tick-AS | Tick-AS-NL | HoD-OA | SSA | HPG |
|---|---|---|---|---|---|---|---|---|---|
| Time (min) | 2 | 4 | 163 | 1 | 40 | 33 | 40 | 37 | 35 |
| Costs (USD) | 0 | 0 | 0 | 0 | 13 | 10 | 2.1 | 1.4 | 1.4 |
| Time (min) | 22 | 8 | 1410 | 2 | 38 | 39 | 128 | 47 | 69 |
| Costs (USD) | 0 | 0 | 0 | 0 | 28 | 18 | 12 | 3.0 | 3.5 |

Victor Zeng and Rakesh M. Verma. Phishbench 2.0: A versatile and extendable benchmarking framework for phishing. In *ACM CODASPY (Demo)*, 2020. doi: 10.1145/3372297.3420017.

Victor Zeng, Shahryar Baki, and Rakesh M. Verma. Diverse datasets and a customizable benchmarking framework for phishing. In *ACM CODASPY*, 2020. doi: 10.1145/3375708.3380313.

Xinnong Zhang, Jiayu Lin, Xinyi Mou, Shiyue Yang, Xiawei Liu, Libo Sun, Hanjia Lyu, Yihang Yang, Weihong Qi, Yue Chen, et al. Socioverse: A world model for social simulation powered by llm agents and a pool of 10 million real-world users. *arXiv preprint arXiv:2504.10157*, 2025.

Yusheng Zhao, Qixin Zhang, Xiao Luo, Weizhi Zhang, Zhiping Xiao, Wei Ju, Philip S Yu, and Ming Zhang. Dynamic text bundling supervision for zero-shot inference on text-attributed graphs. *arXiv preprint arXiv:2505.17599*, 2025.

## A  RELATED WORK

**Traditional simulation methods.** Early approaches to dynamic network simulation often rely on agent-based models (ABMs) (Schelling, 1971; Macal & North, 2009) or event-driven statistical models (Butts, 2008; Blundell et al., 2012). These methods directly encode micro-level behavioral rules or event intensities: for example, ABMs specify decision rules for when agents interact, while REM/Hawkes models use hazard/excitation functions to let past events raise the rate of future ones.

**Dynamic graph generative models.** Two main categories of methods have been proposed to model dynamic graphs: snapshot-based models (Hajiramezanali et al., 2019; Sankar et al., 2020; Pareja et al., 2020) discretize time into snapshots, which coarsens temporal granularity and obscures fine-scale rhythms, while continuous-time methods (Trivedi et al., 2017; 2019; Rossi et al., 2020; Xu et al., 2020; Luo & Li, 2022; 2024) model next edges/events directly but still operate under teacher forcing, predicting the only next edge conditioned on ground-truth history, which emphasizes on short-term predictive accuracy rather than open-loop long-horizon simulation realism.

**LLM multi-agent systems.** Recent work explores LLM-driven agents as simulators of human-like behavior. Task- and game-centric systems (Shen et al., 2023; Yang et al., 2023; FAIR, 2022; Wang et al., 2023) demonstrate planning, role assignment, or collaboration, focusing on game environments including classical normal-form games (Akata et al., 2025; Park et al., 2025), Diplomacy games (FAIR, 2022), and Minecraft (Wang et al., 2023). Sandbox-style systems (Park et al., 2023; Gao et al., 2023; Li et al., 2023; Yang et al., 2024; Piao et al., 2025; Zhang et al., 2025) and opinion-dynamics studies (Chuang et al., 2023; Papachristou & Yuan, 2024; Chang et al., 2025; Ferraro et al., 2024) enable the agents to interact in synthetic environments, demonstrating plausible micro-level behaviors such as conversation, consensus, or triadic closure. Some studies focus on the macro-level

outcomes of LLM multi-agent systems, for instance, consensus v.s. fragmentation in opinion networks (Chuang et al., 2023), overestimation of political homophily (Chang et al., 2025), or diffusion and polarization in online platforms (Yang et al., 2024). Another set of works explores LLM agents for graph generation, mostly targeted at static networks (Zhao et al., 2025; Yu et al., 2025; Du et al., 2025; Ji et al., 2024; Chang et al., 2025; Yao et al., 2024). Although GAG (Ji et al., 2024) moves toward dynamic graph generation, it focuses on actor-item graphs that abstract interactions as consumption/associations rather than direct agent-to-agent exchanges, and its evaluation emphasizes a narrow set of snapshot macro statistics with limited temporal behaviors.

**LLM-based phishing.** The rise of LLMs has significantly escalated phishing threats (Bethany et al., 2024; Heiding et al., 2024). Recent studies show that LLM-crafted phishing emails can easily bypass traditional detection systems, while LLM-based detectors achieve stronger robustness (Afane et al., 2024; Heiding et al., 2024). Hazell (2023) further demonstrate that by retrieving target-specific facts (e.g., scraping public bios) to adapt tone and pretext, LLMs can generate personalized spear-phishing emails with even lower detection rates. Other works explore more comprehensive pipelines: Begou et al. (2023) investigate end-to-end phishing automation with LLMs, capable of generating full phishing kits (site cloning, credential capture, deployment) automatically; Chen et al. (2024) study LLM agents collaborating in specialized roles (e.g., extractor, reasoner) to jailbreak LLMs. Despite these advances, little attention has been paid to how dynamic network structures and multi-node collaboration in email networks would affect phishing effectiveness.

## B   DETAILED DATASET DESCRIPTION & CURATION PROCESS

While both are email corpora, they differ significantly in scale, structure, and conversational style, posing distinct modeling challenges. The Enron dataset is a corporate email corpus from legal proceedings, from which we use a high-activity subset from 2000-2002 containing approximately 34,000 messages from 149 employees. It represents a smaller, organizationally-bounded network with fast-paced, formal exchanges. In contrast, the IETF dataset is constructed from the public email archive of the Internet Engineering Task Force (Internet Engineering Task Force, 2014). Spanning 2015-2025, it includes 165,000 messages from over 1,000 participants across 135 working groups. This network is larger, community-driven, and characterized by more bursty and asynchronous technical discussions. For this dataset, we mapped mailing list aliases to unique agents and preserved cross-postings and time-stamped threads to ensure structural accuracy for long-horizon rollouts.

Finally, for realistic simulations we need agents whose behavior reflects real organizational roles, while prior LLM multi-agent work often differentiates agents only by coarse demographics (e.g., gender, age, occupation) sampled from the population (Park et al., 2023; Yang et al., 2024; Piao et al., 2025). Such simplifications overlook the nuanced social roles, responsibilities, and communication styles that shape behavior in real organizational networks. Therefore, we augment both corpora with role-grounded professional personas for each agent, derived from their actual historical activities. During simulation, these personas guide how agents compose and route messages, thereby helping produce more realistic interactive patterns. The persona construction is guided by GPT-5; more details and examples can be found in Appendix H.1. Table 7 shows that omitting such professional personas significantly reduces simulation fidelity, especially on meso-level interaction patterns characterized by temporal motifs (e.g., thread participation, small-group routines).

In our experiments, for the Enron dataset, we simulate from four start dates {2000-11-27, 2001-04-23, 2001-09-17, 2001-10-22}, each with a 32-day history window. For the IETF dataset, where communications are relatively sparser, we use four start dates {2018-03-01, 2019-11-01, 2022-03-01, 2023-11-01}, each with a 60-day history window.

## C   DETAILED DESCRIPTION OF USED METRICS

We provide detailed definitions of the evaluation metrics. Let $G = (V, E)$ denote a directed network with $|V| = n$ nodes, $|E| = m$ edges, and adjacency matrix $A$. Each edge $e \in E$ has an associated timestamp $t_e$. For time series metrics, let $X = \{X_t\}$ and $Y = \{Y_t\}$ denote simulated and ground-truth activity counts aggregated in 24-hour bins, respectively.

**r24 (Circadian autocorrelation).** We measure the discrepancy in 24-hour lag autocorrelation of hourly activity (restricted to weekdays):

$$\text{r24} = \left| \rho_{24}(X) - \rho_{24}(Y) \right|,$$

where $\rho_{24}(X) = \frac{\text{Cov}(X_t, X_{t+24})}{\sigma(X_t)\,\sigma(X_{t+24})}$.

**HoD (Hour-of-day distribution).** We compare the distributions of activity across the 24 hours of the day. Let $p, q \in \mathbb{R}^{24}$ be normalized histograms of hourly activity from simulation and ground truth. We compute

$$\text{HoD-EMD} = \text{EMD}(p, q),$$

where EMD denotes the Earth Mover's Distance.

**WkndDrop (Weekend activity gap).** We measure the weekend-to-weekday activity ratio:

$$R = \frac{\mu_{\text{weekend}}}{\mu_{\text{weekday}}},$$

where $\mu_{\text{weekend}}$ and $\mu_{\text{weekday}}$ are average counts of emails sent during weekends and weekdays, respectively. The error is

$$\text{WkndDrop} = \left| R^{\text{sim}} - R^{\text{gt}} \right|.$$

**Burstiness (Burst).** For each node $i$, let $\{t_1, \ldots, t_{m_i}\}$ be its sorted timestamps, and define inter-event times $\Delta_k = t_{k+1} - t_k$. The mean and standard deviation are

$$\mu_i = \frac{1}{m_i - 1} \sum_k \Delta_k, \quad \sigma_i = \sqrt{\frac{1}{m_i - 1} \sum_k (\Delta_k - \mu_i)^2}.$$

The burstiness index of node $i$ is

$$B_i = \frac{\sigma_i - \mu_i}{\sigma_i + \mu_i}, \quad B_i \in [-1, 1].$$

We then compare the distributions $\{B_i^{\text{sim}}\}$ and $\{B_i^{\text{gt}}\}$ via

$$\text{Burst-EMD} = \text{EMD}\left( \{B_i^{\text{sim}}\}, \{B_i^{\text{gt}}\} \right).$$

**2-Eg-2h, 2-Eg-8h.** Temporal motif (Paranjape et al., 2017) distributions of 2-edge patterns. We count all ordered pairs of edges $(e_1, e_2)$ with $0 < t_{e_2} - t_{e_1} \leq \Delta$ (with $\Delta$=2h or 8h). Each pair falls into one of six canonical 2-edge motifs:

1. **Reciprocal:** $a \to b$, $b \to a$.
2. **Repeated:** $a \to b$, $a \to b$.
3. **Out-star:** $a \to b$, $a \to c$.
4. **In-star:** $b \to a$, $c \to a$.
5. **Chain-forward:** $a \to b$, $b \to c$.
6. **Chain-backward:** $b \to a$, $c \to b$.

We then compare the motif distributions of simulation and ground truth using Jensen-Shannon Divergence (JSD).

**3-Eg-24h, 3-Eg-48h.** Temporal motif (Paranjape et al., 2017) distributions of 3-edge patterns. We count all ordered triples of edges $(e_1, e_2, e_3)$ with $0 < t_{e_2} - t_{e_1} \leq \Delta$ and $0 < t_{e_3} - t_{e_1} \leq \Delta$ (for $\Delta$=24h or 48h), under strict temporal ordering. We consider the following five 3-edge temporal motifs due to their relevance in email networks:

1. **Dyad Alternation (DYAD_ALT):** $a \to b$, $b \to a$, $a \to b$.
2. **Dyad Burst–Reply (DYAD_BURST_REPLY):** $a \to b$, $a \to b$, $b \to a$.

3. **Feed-forward Closure (FEED_FORWARD_CLOSURE):** $a \to b$, $b \to c$, $a \to c$.

4. **Three-cycle (THREE_CYCLE):** $a \to b$, $b \to c$, $c \to a$.

5. **Broadcast then Cross-link (BROADCAST_THEN_BCorCB):** $a \to b$, $a \to c$, followed by either $b \to c$ or $c \to b$.

We then compare motif distributions between simulation and ground truth using JSD.

**DegDist (Degree distribution).** For each day, let $d_v$ denote the degree of node $v$ in the daily aggregated network. We compute the empirical distribution of $\{d_v : v \in V\}$ for both simulation and ground truth, and measure their discrepancy using EMD. The final score is obtained by averaging the daily EMD values across the evaluation horizon.

**Trans (Transitivity).** We first collapse $G$ into an undirected simple graph $\hat{G}$ (removing edge directions and multi-edges). Transitivity is then defined as

$$\mathrm{Trans}(G) \;=\; \frac{3 \times \#\{\text{triangles in } \hat{G}\}}{\#\{\text{connected triples in } \hat{G}\}}.$$

We compute the daily transitivity values for both simulation and ground truth, and report the root mean squared error (RMSE) across days.

**GlobEff (Global efficiency).** We first collapse $G$ into an undirected simple graph (removing edge directions and multi-edges). Global efficiency is then defined in terms of shortest-path distances $d(u, v)$ between all pairs of distinct nodes:

$$\mathrm{GlobEff}(G) \;=\; \frac{1}{n(n-1)} \sum_{u \neq v} \frac{1}{d(u, v)}.$$

We compute daily values of GlobEff for both simulation and ground truth, and report the RMSE across days.

**Recip (Reciprocity).** Reciprocity is defined as

$$\mathrm{Recip}(G) \;=\; \frac{\left| \{ (u, v) \in E : (v, u) \in E \} \right|}{|E|},$$

i.e., the fraction of edges that are reciprocated. We compute daily reciprocity values for both simulation and ground truth, and report the RMSE across days.

**TopoOvlp (Topology overlap).** We measure the stability of ego-networks across consecutive snapshots. For each node $v$ and consecutive days $t, t+1$, let $N^t(v)$ and $N^{t+1}(v)$ denote its neighbor sets (in- or out-neighbors, without self-loops). The overlap is defined as

$$C_v^t \;=\; \frac{|N^t(v) \cap N^{t+1}(v)|}{\sqrt{|N^t(v)| \cdot |N^{t+1}(v)|}}.$$

For each node $v$, we average $C_v^t$ across all consecutive day-pairs in the horizon. We then compare the distributions of $\{C_v\}$ between simulation and ground truth using EMD.

**DegCen (Degree centrality).** We compute degree centrality for all nodes in each daily snapshot and extract the top-10 nodes by degree. Similarity between simulation and ground truth is measured by the Jaccard index of the two top-10 sets:

$$\mathrm{DegCen} \;=\; \frac{\left| \mathrm{Top}_{10}^{\mathrm{sim}} \cap \mathrm{Top}_{10}^{\mathrm{gt}} \right|}{\left| \mathrm{Top}_{10}^{\mathrm{sim}} \cup \mathrm{Top}_{10}^{\mathrm{gt}} \right|}.$$

We report the average value across days.

**BetwCen (Betweenness centrality).** For each daily snapshot, we compute betweenness centrality for all nodes:

$$\mathrm{BetwCen}(v) \;=\; \sum_{s \neq v \neq t} \frac{\sigma_{st}(v)}{\sigma_{st}},$$

where $\sigma_{st}$ is the number of shortest paths from $s$ to $t$, and $\sigma_{st}(v)$ is the number of those paths passing through $v$. We extract the top-10 nodes ranked by betweenness and measure similarity between simulation and ground truth using the Jaccard index of the two top-10 sets. The final score is the average Jaccard similarity across days.

# D  IMPLEMENTATION DETAILS OF GNNS

Since traditional temporal graph neural networks focus on link prediction given a future time, they do not naturally support network simulation. For each of the following methods, we will first decide on multiple equally distanced future timestamps, and then generate one $n$-by-$n$ matrix where $n$ is the number of nodes for each timestamp, that indicates the predicted probabilities where there exist directed edges from the nodes with the row indices as the ID to the nodes with the column indices as the ID. Given these probability matrices, we will determine thresholds such that we keep the edges where the probabilities are above the thresholds as the simulated edges.

We provide two variants on determining the thresholds. **Approach 1:** We evaluate the edge density of the temporal graph snapshot between the first and the second future timestamps and then determine the threshold such that the number of simulated edges matches that density. Then, we keep on using this threshold throughout the simulation for all future timestamps. **Approach 2:** We evaluate the edge density of every temporal graph snapshot between the consecutive timestamps and control the threshold *dynamically* such that the number of simulated edges of each timestamp matches the density of each temporal graph snapshot. We have tried using a threshold estimated based on the training and validation sets. However, the threshold that works well for training and validation can become an extreme threshold, such that in the worst-case, no edges can remain after the cutoff.

We evaluate both tuning approaches for all dynamic GNNs, and report results using **Approach 2**, which consistently performs better. Next, we introduce the detailed steps to construct the probability matrices for each of the baselines. We note that for each of these baselines, we tune the model hyperparameters based on the link-prediction performances on the validation sets.

**DySAT (Sankar et al., 2020).**[1] DySAT is a snapshot-based approach for temporal link prediction. It assumes that the time span between every consecutive snapshots is equal. It keeps track of the up-to-date node embeddings, which capture the contexts of all past graph snapshots via dynamic self-attention. Given a pair of node embeddings $(Z_u, Z_v)$, we perform the following operation to construct the embedding of the directed edge $u \rightarrow v$: $\ln Z_u^{|\cdot|} \otimes Z_u \otimes Z_v$, where $Z_u^{|\cdot|}$ denotes the element-wise absolute value of $Z_u$ and $\otimes$ denotes element-wise multiplication. Then, a logistic regression model is trained on a list of true edges with positive labels from the training set and a list of non-edges with negative labels. For edge simulation, we first construct the edge embeddings for every ordered pair of node embeddings. Then we pass the edge embeddings to the logistic regression model to get the predicted probabilities that the edges are true. Note that our approach of constructing edge embeddings is different from the original approach because the simulated edges we are interested in should be directed, while the original approach that adopts the Hadammad product omits the order of the nodes. We have verified that the change improves the prediction performance on directed edges compared to the original approach.

**EvolveGCN (Pareja et al., 2020).**[2] EvolveGCN is also a snapshot-based approach with the assumption that the snapshots are equally spaced in time. It adapts the graph convolutional network (GCN (Kipf & Welling, 2017)) model along the temporal dimension and captures the dynamism of the graph sequence via an RNN to evolve the GCN parameters. Similar to DySAT, for each of the future timestamps we are interested in, we pass the historical information into the model to compute all node embeddings of the future snapshots. The link predictor adopted by EvolveGCN first inputs a pair of node embeddings to a single-layer perceptron and then applies the softmax function on the output of the perceptron. Using the same link predictor, we gather the predicted probabilities of the existence of edges for all ordered node pairs.

**NLB (Luo & Li, 2024).**[3] NLB is a more recent approach in temporal graph representation learning (TGRL) that directly processes a stream of temporal links rather than snapshots with timestamps. It avoids backtracking in time to sample historical interaction, as commonly performed by the previous literature in TGRL, with its proposed GPU-compatible forward-sampling. Thus, it is highly scalable

---

[1]For DySAT, we adapt the implementation in `https://github.com/FeiGSSS/DySAT_pytorch/tree/main` for network simulation.

[2]For EvolveGCN, we adapt the implementation in `https://github.com/IBM/EvolveGCN` for network simulation.

[3]For NLB, we adapt the implementation in `https://github.com/Graph-COM/NLB` for network simulation.

Table 7: Ablation study on the augmented professional persona, using Enron dataset as an example. **Bold** indicates the best rank across each metric category.

| Model | Temporal Rhythms | | | | | Temporal Dynamics | | | | | Global Topology | | | | | Local Topology | | | |
|---|---|---|---|---|---|---|---|---|---|---|---|---|---|---|---|---|---|---|---|
| | r24 | HoD | WkndDrop | Burst | Rank | 2-Eg-2h | 2-Eg-8h | 3-Eg-24h | 3-Eg-48h | Rank | DegDist | Trans | GlobEff | Recip | Rank | TopoOvlp | DegCen↑ | BetwCen↑ | Rank |
| HPG-NoProPersona | 0.15 | 1.00 | 0.13 | 0.14 | 1.8 | 0.35 | 0.35 | 0.41 | 0.43 | 2.0 | 0.28 | 0.16 | 0.012 | 0.11 | 1.5 | 0.07 | 0.28 | 0.56 | 1.3 |
| HPG | 0.15 | 0.77 | 0.05 | 0.13 | **1.0** | 0.33 | 0.28 | 0.27 | 0.25 | **1.0** | 0.25 | 0.19 | 0.011 | 0.11 | **1.3** | 0.06 | 0.28 | 0.56 | **1.0** |

and suitable for adaptation to simulate large-scale network evolutions. For both datasets we used, it can efficiently handle the querying of all ordered node pairs in one batched call. For each timestamp we perform the simulation, we query NLB for all temporal node embeddings and then adopt the Multi-layer-Perceptron-based link predictor used by NLB to get a predicted probability of edge existence for all ordered node pairs. As NLB converts the timestamp into temporal features, we also pass the timestamps used for simulation into the model.

## E    DETAILED RESULTS OF SENSITIVITY STUDIES

Tables 8, 9, and 10 show the raw results of our sensitivity studies. They provide the raw data to plot Figure 2.

## F    SUPPLEMENTARY SIMULATION RESULTS

Table 6 reports the average runtime and cost of each method, computed over four 8-day simulations on two datasets. For non-LLM methods, all experiments were run locally (using NVIDIA RTX 6000 Ada GPUs when possible) and therefore incur no LLM-related costs; among them, only EvolveGCN is inefficient, while the others run very quickly. For LLM-based methods, runtimes are broadly comparable, though some variability may arise from fluctuations in OpenAI's API response time. Costs, however, differ: methods that rely on fixed timesteps and query LLM agents multiple times per day incur substantial expenses, whereas those with more efficient scheduling keep costs manageable.

To evaluate how similar the LLM-agent-generated content is to real content, we adopt standard metrics from the authorship retrieval literature, including Mean Reciprocal Rank (MRR), Hit@1, and Hit@5. We compute embeddings for both generated and real content, then construct similarity matrices across all agents. Hit@1 measures the likelihood that a generated sample is most similar to the corresponding agent's real content, while Hit@5 relaxes this criterion to the top five matches. MRR provides a ranking-based measure of retrieval quality.

We consider three embedding approaches: (1) TF-IDF, which emphasizes word-level similarity; (2) Jaccard similarity, which captures topical overlap; and (3) OpenAI's embedding model (OpenAI, 2024), which encodes semantic information in vector space. The results are summarized in Table 5, where we report HPG's performance as a demonstration, since all LLM-based methods rely on the same backend model and use similar prompts.

Overall, the LLM agents generate content that is stylistically similar to real authors. On Enron, the generated content achieves great similarity, with the correct author appearing in the top five about two-thirds of the time. On IETF, performance is stronger, reaching up to 60% Hit@1, indicating that the agents more consistently reproduce the distinctive writing styles of individual participants.

## G    SUPPLEMENTARY PHISHING SYNTHESIS RESULTS

Table 11 reports complementary metrics to the main results in Table 4, using a fixed threshold of 0.5 for probabilistic outputs. Notably, BERT-A (Dahal, 2025) and BERT-B (Cheptoo, 2024) achieve over 96% and 99% accuracy, respectively, on traditional phishing datasets, yet these benchmarks appear ill-suited for evaluating advanced context-aware attacks, where these models would collapse. Combined with the false positive rates (FPR) reported in Table 12, we observe that all methods except Glimpse maintain reasonable FPR on legitimate emails. Glimpse, however, tends to misclassify a substantial number of legitimate emails as AI-generated, suggesting limited generalization ability.

Table 8: Raw results of the sensitivity study on trigger node ratios. Enron (top) and IETF (bottom).

| Triggers | Model | Temporal Rhythms | | | | Temporal Dynamics | | | | Global Topology | | | | Local Topology | | |
|---|---|---|---|---|---|---|---|---|---|---|---|---|---|---|---|---|
| | | r24 | HoD | WkndDrop | Burst | 2-Eg-2h | 2-Eg-8h | 3-Eg-24h | 3-Eg-48h | DegDist | Trans | GlobEff | Recip | TopoOvlp | DegCen↑ | BetwCen↑ |
| T=0% | Hawkes | 0.23 | 0.86 | 0.095 | 0.22 | 0.18 | 0.11 | 0.2 | 0.16 | 0.36 | 0.14 | 0.019 | 0.12 | 0.037 | 0.28 | 0.34 |
| | NLB | 0.2 | 1.9 | 0.58 | 0.21 | 0.33 | 0.29 | 0.46 | 0.42 | 0.71 | 0.35 | 0.037 | 0.15 | 0.1 | 0.1 | 0.2 |
| | HPG | 0.37 | 1.2 | 0.17 | 0.35 | 0.43 | 0.45 | 0.48 | 0.48 | 0.8 | 0.25 | 0.046 | 0.32 | 0.22 | 0.2 | 0.28 |
| T=5% | Hawkes | 0.14 | 0.95 | 0.036 | 0.21 | 0.26 | 0.16 | 0.26 | 0.17 | 0.23 | 0.16 | 0.014 | 0.12 | 0.025 | 0.29 | 0.42 |
| | NLB | 0.4 | 1.7 | 0.59 | 0.36 | 0.44 | 0.4 | 0.46 | 0.41 | 0.8 | 0.37 | 0.025 | 0.21 | 0.22 | 0.1 | 0.33 |
| | HPG | 0.23 | 1.2 | 0.076 | 0.29 | 0.29 | 0.24 | 0.26 | 0.24 | 0.37 | 0.22 | 0.021 | 0.13 | 0.12 | 0.27 | 0.45 |
| T=10% | Hawkes | 0.08 | 0.74 | 0.02 | 0.16 | 0.35 | 0.24 | 0.33 | 0.25 | 0.16 | 0.13 | 0.0130 | 0.09 | 0.02 | 0.29 | 0.48 |
| | NLB | 0.32 | 1.8 | 0.52 | 0.2 | 0.34 | 0.27 | 0.39 | 0.34 | 0.69 | 0.37 | 0.037 | 0.21 | 0.086 | 0.075 | 0.35 |
| | HPG | 0.15 | 0.77 | 0.05 | 0.13 | 0.33 | 0.28 | 0.27 | 0.25 | 0.25 | 0.19 | 0.0113 | 0.11 | 0.06 | 0.28 | 0.56 |
| T=20% | Hawkes | 0.24 | 1 | 0.039 | 0.15 | 0.39 | 0.29 | 0.35 | 0.33 | 0.13 | 0.11 | 0.0068 | 0.1 | 0.023 | 0.31 | 0.61 |
| | NLB | 0.46 | 2.1 | 0.46 | 0.32 | 0.45 | 0.33 | 0.6 | 0.49 | 0.53 | 0.26 | 0.029 | 0.12 | 0.13 | 0.16 | 0.45 |
| | HPG | 0.26 | 1.3 | 0.046 | 0.2 | 0.35 | 0.33 | 0.47 | 0.42 | 0.17 | 0.28 | 0.0044 | 0.12 | 0.045 | 0.32 | 0.73 |
| T=0% | Hawkes | 0.048 | 1.1 | 0.2 | 0.35 | 0.17 | 0.16 | 0.24 | 0.18 | 0.13 | 0.32 | 0.00094 | 0.18 | 0.046 | 0.076 | 0.4 |
| | NLB | 0.32 | 2.5 | 0.32 | 0.45 | 0.49 | 0.43 | 0.32 | 0.3 | 0.18 | 0.35 | 0.0057 | 0.23 | 0.05 | 0.051 | 0.47 |
| | HPG | 0.29 | 2.1 | 0.31 | 0.24 | 0.19 | 0.16 | 0.39 | 0.29 | 0.15 | 0.32 | 0.0014 | 0.21 | 0.11 | 0.091 | 0.52 |
| T=5% | Hawkes | 0.18 | 1 | 0.11 | 0.21 | 0.25 | 0.19 | 0.44 | 0.37 | 0.066 | 0.24 | 0.00075 | 0.14 | 0.018 | 0.091 | 0.5 |
| | NLB | 0.45 | 1.4 | 0.37 | 0.32 | 0.39 | 0.3 | 0.42 | 0.44 | 0.21 | 0.41 | 0.0028 | 0.26 | 0.19 | 0.12 | 0.67 |
| | HPG | 0.21 | 1.2 | 0.29 | 0.31 | 0.39 | 0.21 | 0.35 | 0.31 | 0.096 | 0.2 | 0.00085 | 0.18 | 0.058 | 0.14 | 0.76 |
| T=10% | Hawkes | 0.19 | 1.1 | 0.11 | 0.14 | 0.44 | 0.32 | 0.51 | 0.35 | 0.06 | 0.20 | 0.0008 | 0.13 | 0.01 | 0.10 | 0.56 |
| | NLB | 0.22 | 1.5 | 0.35 | 0.48 | 0.46 | 0.37 | 0.44 | 0.53 | 0.32 | 0.39 | 0.013 | 0.23 | 0.28 | 0.1 | 0.66 |
| | HPG | 0.32 | 1.7 | 0.27 | 0.58 | 0.36 | 0.31 | 0.38 | 0.35 | 0.11 | 0.31 | 0.0009 | 0.19 | 0.07 | 0.18 | 0.82 |
| T=20% | Hawkes | 0.12 | 1.8 | 0.3 | 0.42 | 0.48 | 0.4 | 0.25 | 0.36 | 0.067 | 0.14 | 0.00069 | 0.094 | 0.0088 | 0.13 | 0.73 |
| | NLB | 0.36 | 1.6 | 0.58 | 0.43 | 0.56 | 0.52 | 0.47 | 0.48 | 0.2 | 0.38 | 0.0037 | 0.14 | 0.13 | 0.21 | 0.72 |
| | HPG | 0.3 | 1.8 | 0.57 | 0.6 | 0.39 | 0.38 | 0.28 | 0.34 | 0.097 | 0.23 | 0.001 | 0.12 | 0.061 | 0.21 | 0.92 |

Table 9: Raw results of the sensitivity study on history length. Enron (top) and IETF (bottom).

| History | Model | Temporal Rhythms | | | | Temporal Dynamics | | | | Global Topology | | | | Local Topology | | |
|---|---|---|---|---|---|---|---|---|---|---|---|---|---|---|---|---|
| | | r24 | HoD | WkndDrop | Burst | 2-Eg-2h | 2-Eg-8h | 3-Eg-24h | 3-Eg-48h | DegDist | Trans | GlobEff | Recip | TopoOvlp | DegCen↑ | BetwCen↑ |
| H=8 | Hawkes | 0.13 | 1.6 | 4.3 | 0.36 | 0.37 | 0.32 | 0.29 | 0.3 | 0.76 | 0.18 | 0.063 | 0.11 | 0.038 | 0.26 | 0.54 |
| | NLB | 0.36 | 1.7 | 0.4 | 0.13 | 0.36 | 0.27 | 0.38 | 0.33 | 0.47 | 0.23 | 0.044 | 0.2 | 0.047 | 0.081 | 0.29 |
| | HPG | 0.3 | 2.9 | 2.8 | 0.17 | 0.35 | 0.3 | 0.3 | 0.27 | 0.64 | 0.26 | 0.014 | 0.13 | 0.088 | 0.26 | 0.53 |
| H=16 | Hawkes | 0.059 | 0.67 | 0.07 | 0.19 | 0.29 | 0.27 | 0.41 | 0.44 | 0.18 | 0.15 | 0.0091 | 0.11 | 0.024 | 0.28 | 0.55 |
| | NLB | 0.19 | 1.9 | 0.54 | 0.26 | 0.42 | 0.32 | 0.39 | 0.35 | 0.58 | 0.27 | 0.043 | 0.17 | 0.074 | 0.069 | 0.39 |
| | HPG | 0.28 | 1.6 | 0.043 | 0.2 | 0.34 | 0.29 | 0.37 | 0.34 | 0.31 | 0.25 | 0.011 | 0.11 | 0.069 | 0.29 | 0.57 |
| H=32 | Hawkes | 0.08 | 0.74 | 0.02 | 0.16 | 0.35 | 0.24 | 0.33 | 0.25 | 0.16 | 0.13 | 0.0130 | 0.09 | 0.02 | 0.29 | 0.48 |
| | NLB | 0.32 | 1.8 | 0.52 | 0.2 | 0.34 | 0.27 | 0.39 | 0.34 | 0.69 | 0.37 | 0.037 | 0.21 | 0.086 | 0.075 | 0.35 |
| | HPG | 0.15 | 0.77 | 0.05 | 0.13 | 0.33 | 0.28 | 0.27 | 0.25 | 0.25 | 0.19 | 0.0113 | 0.11 | 0.06 | 0.28 | 0.56 |
| H=64 | Hawkes | 0.1 | 1.1 | 0.041 | 0.19 | 0.32 | 0.21 | 0.31 | 0.27 | 0.18 | 0.14 | 0.0099 | 0.09 | 0.025 | 0.31 | 0.49 |
| | NLB | 0.36 | 1.6 | 0.41 | 0.24 | 0.3 | 0.28 | 0.38 | 0.35 | 0.57 | 0.45 | 0.033 | 0.2 | 0.13 | 0.1 | 0.4 |
| | HPG | 0.13 | 0.89 | 0.065 | 0.16 | 0.29 | 0.25 | 0.33 | 0.33 | 0.31 | 0.24 | 0.011 | 0.14 | 0.071 | 0.29 | 0.54 |
| H=15 | Hawkes | 0.046 | 1.5 | 0.11 | 0.2 | 0.45 | 0.36 | 0.42 | 0.42 | 0.065 | 0.28 | 0.00078 | 0.13 | 0.013 | 0.13 | 0.62 |
| | NLB | 0.44 | 1.1 | 0.39 | 0.55 | 0.42 | 0.33 | 0.52 | 0.52 | 0.14 | 0.26 | 0.0036 | 0.12 | 0.027 | 0.043 | 0.46 |
| | HPG | 0.28 | 2 | 0.26 | 0.5 | 0.41 | 0.39 | 0.33 | 0.35 | 0.083 | 0.25 | 0.00058 | 0.15 | 0.048 | 0.19 | 0.89 |
| H=30 | Hawkes | 0.18 | 1 | 0.14 | 0.15 | 0.41 | 0.33 | 0.43 | 0.32 | 0.066 | 0.24 | 0.00096 | 0.12 | 0.012 | 0.11 | 0.59 |
| | NLB | 0.21 | 1.1 | 0.46 | 0.46 | 0.38 | 0.26 | 0.4 | 0.41 | 0.15 | 0.35 | 0.0028 | 0.17 | 0.052 | 0.11 | 0.54 |
| | HPG | 0.26 | 1.3 | 0.22 | 0.49 | 0.35 | 0.33 | 0.33 | 0.23 | 0.083 | 0.19 | 0.00071 | 0.16 | 0.041 | 0.15 | 0.79 |
| H=60 | Hawkes | 0.19 | 1.1 | 0.11 | 0.14 | 0.44 | 0.32 | 0.51 | 0.35 | 0.06 | 0.20 | 0.0008 | 0.13 | 0.01 | 0.10 | 0.56 |
| | NLB | 0.22 | 1.5 | 0.35 | 0.48 | 0.46 | 0.37 | 0.44 | 0.53 | 0.32 | 0.39 | 0.013 | 0.23 | 0.28 | 0.1 | 0.66 |
| | HPG | 0.32 | 1.7 | 0.27 | 0.58 | 0.36 | 0.31 | 0.38 | 0.35 | 0.11 | 0.31 | 0.0009 | 0.19 | 0.07 | 0.18 | 0.82 |
| H=120 | Hawkes | 0.14 | 1.4 | 0.18 | 0.12 | 0.36 | 0.31 | 0.25 | 0.25 | 0.055 | 0.29 | 0.0006 | 0.14 | 0.017 | 0.071 | 0.66 |
| | NLB | 0.44 | 1.3 | 0.41 | 0.5 | 0.42 | 0.37 | 0.51 | 0.52 | 0.25 | 0.51 | 0.0021 | 0.35 | 0.27 | 0.13 | 0.78 |
| | HPG | 0.43 | 2.2 | 0.32 | 0.55 | 0.45 | 0.29 | 0.38 | 0.36 | 0.088 | 0.18 | 0.00081 | 0.19 | 0.039 | 0.14 | 0.73 |

We further evaluate CA-LLM under varying amounts of accessible email history (Table 13). Increasing the number of past emails generally improves detection, but the gains diminish once too much history is added. This is intuitive, as only recent communications are most relevant for capturing the temporal context of email interactions.

# H  PROMPTS & EXAMPLE RESPONSES

## H.1  PROMPTS FOR DATASET CURATION

Figure 3 presents the prompts used to summarize the professional personas of employees or participants in the Enron and IETF datasets. Figure 4 provides an example output for an IETF participant, illustrating how the prompt captures key aspects of their role, expertise, and communication style.

## H.2  PROMPTS FOR NETWORK SIMULATION

Figure 5 presents the prompts used for HPG. Other LLM-based agent systems employ highly similar prompts, with only minor adjustments to account for different settings such as activation schedules.

Table 10: Raw results of the sensitivity study on simulation horizon. Enron (top) and IETF (bottom).

| Horizon | Model | Temporal Rhythms | | | | Temporal Dynamics | | | | Global Topology | | | | Local Topology | | |
|---|---|---|---|---|---|---|---|---|---|---|---|---|---|---|---|---|
| | | r24 | HoD | WkndDrop | Burst | 2-Eg-2h | 2-Eg-8h | 3-Eg-24h | 3-Eg-48h | DegDist | Trans | GlobEff | Recip | TopoOvlp | DegCen ↑ | BetwCen ↑ |
| D=8 | Hawkes | 0.08 | 0.74 | 0.02 | 0.16 | 0.35 | 0.24 | 0.33 | 0.25 | 0.16 | 0.13 | 0.0130 | 0.09 | 0.02 | 0.29 | 0.48 |
| | NLB | 0.32 | 1.8 | 0.52 | 0.2 | 0.34 | 0.27 | 0.39 | 0.34 | 0.69 | 0.37 | 0.037 | 0.21 | 0.086 | 0.075 | 0.35 |
| | HPG | 0.15 | 0.77 | 0.05 | 0.13 | 0.33 | 0.28 | 0.27 | 0.25 | 0.25 | 0.19 | 0.0113 | 0.11 | 0.06 | 0.28 | 0.56 |
| D=16 | Hawkes | 0.18 | 0.73 | 0.062 | 0.21 | 0.24 | 0.17 | 0.21 | 0.19 | 0.17 | 0.16 | 0.01 | 0.15 | 0.022 | 0.29 | 0.43 |
| | NLB | 0.27 | 1.6 | 0.46 | 0.28 | 0.37 | 0.3 | 0.52 | 0.46 | 0.63 | 0.29 | 0.032 | 0.15 | 0.079 | 0.065 | 0.31 |
| | HPG | 0.22 | 0.59 | 0.043 | 0.099 | 0.37 | 0.36 | 0.34 | 0.29 | 0.29 | 0.21 | 0.016 | 0.18 | 0.043 | 0.29 | 0.51 |
| D=24 | Hawkes | 0.13 | 0.64 | 0.036 | 0.18 | 0.23 | 0.18 | 0.36 | 0.31 | 0.19 | 0.15 | 0.015 | 0.16 | 0.023 | 0.28 | 0.45 |
| | NLB | 0.2 | 1.8 | 0.51 | 0.2 | 0.38 | 0.32 | 0.56 | 0.52 | 0.49 | 0.25 | 0.039 | 0.21 | 0.04 | 0.084 | 0.34 |
| | HPG | 0.23 | 0.72 | 0.07 | 0.15 | 0.31 | 0.32 | 0.26 | 0.24 | 0.28 | 0.2 | 0.02 | 0.18 | 0.042 | 0.28 | 0.52 |
| D=32 | Hawkes | 0.094 | 0.66 | 0.028 | 0.22 | 0.22 | 0.16 | 0.37 | 0.3 | 0.19 | 0.16 | 0.015 | 0.18 | 0.024 | 0.27 | 0.44 |
| | NLB | 0.26 | 1.9 | 0.52 | 0.26 | 0.24 | 0.2 | 0.49 | 0.42 | 0.57 | 0.46 | 0.027 | 0.23 | 0.14 | 0.12 | 0.38 |
| | HPG | 0.2 | 0.7 | 0.056 | 0.15 | 0.28 | 0.29 | 0.31 | 0.29 | 0.28 | 0.19 | 0.019 | 0.18 | 0.049 | 0.3 | 0.52 |
| D=8 | Hawkes | 0.19 | 1.1 | 0.11 | 0.14 | 0.44 | 0.32 | 0.51 | 0.35 | 0.06 | 0.20 | 0.0008 | 0.13 | 0.01 | 0.10 | 0.56 |
| | NLB | 0.22 | 1.5 | 0.35 | 0.48 | 0.46 | 0.37 | 0.44 | 0.53 | 0.32 | 0.39 | 0.013 | 0.23 | 0.28 | 0.1 | 0.66 |
| | HPG | 0.32 | 1.7 | 0.27 | 0.58 | 0.36 | 0.31 | 0.38 | 0.35 | 0.11 | 0.31 | 0.0009 | 0.19 | 0.07 | 0.18 | 0.82 |
| D=16 | Hawkes | 0.12 | 1.2 | 0.15 | 0.15 | 0.29 | 0.21 | 0.44 | 0.44 | 0.061 | 0.26 | 0.00062 | 0.13 | 0.015 | 0.13 | 0.66 |
| | NLB | 0.43 | 1.3 | 0.31 | 0.22 | 0.48 | 0.42 | 0.38 | 0.38 | 0.24 | 0.41 | 0.007 | 0.35 | 0.16 | 0.081 | 0.85 |
| | HPG | 0.1 | 1.3 | 0.18 | 0.29 | 0.24 | 0.2 | 0.39 | 0.36 | 0.074 | 0.27 | 0.0007 | 0.13 | 0.036 | 0.17 | 0.86 |
| D=24 | Hawkes | 0.094 | 1.2 | 0.13 | 0.19 | 0.29 | 0.2 | 0.37 | 0.36 | 0.062 | 0.23 | 0.00057 | 0.12 | 0.02 | 0.1 | 0.65 |
| | NLB | 0.35 | 1.2 | 0.39 | 0.41 | 0.3 | 0.24 | 0.52 | 0.52 | 0.23 | 0.65 | 0.0075 | 0.36 | 0.13 | 0.17 | 0.74 |
| | HPG | 0.11 | 1.4 | 0.18 | 0.23 | 0.23 | 0.19 | 0.44 | 0.42 | 0.084 | 0.26 | 0.00074 | 0.13 | 0.024 | 0.15 | 0.84 |
| D=32 | Hawkes | 0.052 | 0.92 | 0.1 | 0.23 | 0.23 | 0.18 | 0.4 | 0.33 | 0.062 | 0.24 | 0.00056 | 0.12 | 0.024 | 0.11 | 0.67 |
| | NLB | 0.35 | 1.3 | 0.37 | 0.3 | 0.48 | 0.44 | 0.36 | 0.32 | 0.22 | 0.45 | 0.0093 | 0.22 | 0.18 | 0.11 | 0.79 |
| | HPG | 0.1 | 1.3 | 0.11 | 0.22 | 0.19 | 0.15 | 0.37 | 0.29 | 0.11 | 0.26 | 0.0013 | 0.13 | 0.028 | 0.14 | 0.81 |

Table 11: Detection results on Enron for synthesized phishing emails (FNR, Precision, and F1) across attacker settings.

| | | Single-Node | | | | | Multi-Node (Info Sharing) | | | | Multi-Node (Node Collab) | | | |
|---|---|---|---|---|---|---|---|---|---|---|---|---|---|---|
| | Context | ✗ | ✗ | ✓ | ✓ | ✓ | ✓ | ✓ | ✓ | ✓ | ✓ | ✓ | ✓ | ✓ |
| | Targeting | Random | Top | Self | Random | Top | Top | Top | Top | Top | Top | Top | Top | Top |
| | # Attackers | 1 | 1 | 1 | 1 | 1 | 2 | 3 | 4 | 5 | 2 | 3 | 4 | 5 |
| CA-LLM | FNR | 0.220 | 0.210 | 0.220 | 0.333 | 0.384 | 0.541 | 0.592 | 0.579 | 0.561 | 0.750 | 0.670 | 0.550 | 0.650 |
| | Precision | 0.587 | 0.590 | 0.587 | 0.546 | 0.526 | 0.450 | 0.421 | 0.421 | 0.439 | 0.313 | 0.375 | 0.450 | 0.389 |
| | F1 | 0.670 | 0.675 | 0.670 | 0.600 | 0.567 | 0.455 | 0.415 | 0.421 | 0.439 | 0.278 | 0.351 | 0.450 | 0.368 |
| Glimpse | FNR | 0.530 | 0.560 | 0.510 | 0.816 | 0.939 | 0.888 | 0.830 | 0.839 | 0.869 | 0.840 | 0.820 | 0.890 | 0.840 |
| | Precision | 0.141 | 0.133 | 0.146 | 0.059 | 0.021 | 0.037 | 0.056 | 0.050 | 0.043 | 0.053 | 0.059 | 0.037 | 0.053 |
| | F1 | 0.217 | 0.205 | 0.225 | 0.090 | 0.031 | 0.056 | 0.084 | 0.076 | 0.065 | 0.080 | 0.089 | 0.055 | 0.080 |
| BERT-A | FNR | 0.890 | 0.900 | 0.670 | 0.980 | 1.000 | 1.000 | 1.000 | 0.990 | 0.990 | 1.000 | 1.000 | 1.000 | 1.000 |
| | Precision | 0.917 | 0.909 | 0.971 | 0.667 | 0.000 | 0.000 | 0.000 | 0.500 | 0.500 | 0.000 | 0.000 | 0.000 | 0.000 |
| | F1 | 0.196 | 0.180 | 0.493 | 0.039 | 0.000 | 0.000 | 0.000 | 0.020 | 0.020 | 0.000 | 0.000 | 0.000 | 0.000 |
| BERT-B | FNR | 0.850 | 0.830 | 0.650 | 0.990 | 0.950 | 0.979 | 1.000 | 0.979 | 0.969 | 1.000 | 1.000 | 0.990 | 0.990 |
| | Precision | 1.000 | 1.000 | 1.000 | 1.000 | 1.000 | 1.000 | 0.000 | 1.000 | 1.000 | 0.000 | 0.000 | 1.000 | 1.000 |
| | F1 | 0.261 | 0.291 | 0.519 | 0.020 | 0.095 | 0.040 | 0.000 | 0.040 | 0.059 | 0.000 | 0.000 | 0.020 | 0.020 |

Table 12: False Positive Rate (FPR) by defender model on Enron.

| Defender Model | FPR |
|---|---|
| Glimpse | 0.2205 |
| BERT-A | 0.0008 |
| BERT-B | 0.0000 |
| CA-LLM | 0.0424 |

Table 13: Detection performance vs. the number of history emails accessible to context-aware attacker when attacking against the CA-LLM denfender.

| # Emails | AUC | Precision | Recall | F1 | FPR | FNR |
|---|---|---|---|---|---|---|
| 1 | 0.7761 | 0.4918 | 0.6000 | 0.5405 | 0.0478 | 0.4000 |
| 5 | 0.7311 | 0.4513 | 0.5100 | 0.4789 | 0.0478 | 0.4900 |
| 10 | 0.7337 | 0.4513 | 0.5152 | 0.4811 | 0.0478 | 0.4848 |
| 20 | 0.7461 | 0.4655 | 0.5400 | 0.5000 | 0.0478 | 0.4600 |

---

**Prompts for Summarizing Employees' Professional Personas**

**SYSTEM:**
You are a specialized text analysis assistant. You have access to the full email history of {employee_name}. Your goal is to analyze this email history and extract relevant information that can inform how to realistically simulate this employee's behavior in future email interactions.

---

**USER:**
Please analyze the following email history carefully. Then provide a concise but detailed summary of the employee's key attributes:

1. Role & Responsibilities:
• Summarize the employee's job title or primary role within the organization.
• List any core duties, responsibilities, or specific tasks they are known to handle.

2. Topics & Domains of Expertise:
• Identify main areas of knowledge, expertise, or recurring subject matter they discuss.
• Note any specialized jargon, technical skills, or industry-specific knowledge.

3. Interaction Patterns:
• Describe how this employee typically communicates with others (tone, style, formality, etc.).
• What are the patterns of initiating or responding to emails (e.g., rarely initiates, often loops in certain teams, etc.)?

4. Decision-Making & Authority Level:
• Indicate whether the employee has decision-making power in certain areas.
• Identify if they approve/deny requests, give instructions, or escalate issues to higher management.

5. Response Behavior:
• Observe how promptly they respond.
• Note whether they provide detailed answers, short confirmations, or request clarifications frequently.

6. Key Partnerships & Collaboration:
• Identify who they regularly interact with.
• Mention any cross-departmental communication patterns or recurring collaborations.

7. Notable Projects & Achievements:
• Highlight major projects, accomplishments, and milestones referenced in their emails.

8. Communication Nuances:
• Note any unique habits or quirks, such as consistent sign-offs, emoticon usage, or certain phrases.

9. Potential Motivations or Concerns:
• Mention if they appear driven by specific goals (e.g., career growth, innovation, competition, etc.).
• Identify any concerns or pain points that frequently appear.

10. Overall Summary:
• Provide an at-a-glance overview of the employee's professional profile.
• Include any additional points that would be relevant to simulating their email style or decision-making.

# Email History to Analyze
""" {email_history} """

# Output Format & Style
• Present your findings clearly, using the headings outlined above.
• Focus on accuracy and clarity, using the emails as evidence.
• If a particular detail is unclear or contradictory, note your uncertainty.

Figure 3: Prompts for Summarizing Employees' Professional Personas.

## Example LLM Analysis Response

**LLM RESPONSE:**

# 1. Role & Responsibilities
- **Job Title**: Employee A is involved primarily with the Internet Engineering Task Force (IETF) in a senior engineering role, likely as a chair or influential member of several working groups focusing on networking protocols and standards.
- **Core Duties**: Their responsibilities encompass reviewing and endorsing drafts, conducting discussions on various networking protocols (especially BGP, MPLS, and EVPN), and overseeing technical direction for projects within community discussions. They are also engaged in organizing workshops related to network programming, particularly with the P4 language.

# 2. Topics & Domains of Expertise
- **Areas of Knowledge**: Employee A's expertise spans networking technologies, focusing on protocols like BGP, MPLS, EVPN, P4 programming, and related network architectures (SmartNICs, ASICs). Discussions often cover operational challenges, networking security, and advancements in network programmability.
- **Specialized Jargon**: They frequently use technical terms such as "BFD," "control packets," "AS-Specific," "UDP source port," "P4 APIs," and "Geneve," demonstrating fluency in networking and P4-related terminology.

# 3. Interaction Patterns
- **Communication Style**: Communicates in a friendly yet professional manner, using casual sign-offs like "Cheers" or "Best regards." Maintains a collegial atmosphere while being assertive and exhibiting technical depth in interactions.
- **Email Behavior**: Typically responds quickly to inquiries, contributing actively to conversations and often including many recipients. While engaged in discussions initiated by others, also initiates topics when necessary, particularly around technical issues.

# 4. Decision-Making & Authority Level
- **Authority Level**: Possesses significant decision-making authority within technical discussions, acting as a consultant or advisor for networking standards and protocols. Engagement includes approving drafts and providing critical feedback on proposals.
- **Role in Approvals**: Actively involved in the review processes integral to the IETF's working groups, influencing decisions related to draft adoption and updates.

# 5. Response Behavior
- **Response Promptness**: Known for prompt responses, often replying the same day or within a few hours.
- **Details in Responses**: Correspondence usually contains in-depth analyses or clarifications rather than simplistic affirmations, reflecting extensive knowledge and engagement.

# 6. Key Partnerships & Collaboration
- **Regular Interactions**: Collaborates with numerous professionals across IETF working groups (e.g., BESS, RTGWG, MPLS).
- **Cross-Departmental Communication**: Communication often extends across multiple organizations and teams involved in networking standards, highlighting a collaborative approach.

# 7. Notable Projects & Achievements
- **Achievements**: Noteworthy contributions include significant drafts related to the development of network protocols, especially those improving routing and programmability. Involvement in organizing workshops demonstrates leadership in fostering community awareness around P4 programming and related innovations.

# 8. Communication Nuances
- **Unique Habits**: Often uses informal yet engaging sign-offs and incorporates technical analogies humorously. Ability to connect technical discussions with accessible language enhances effectiveness.

# 9. Potential Motivations or Concerns
- **Motivations**: Driven by a passion for advancing networking technology, committed to enhancing industry standards and improving networking practices through collaboration and open communication.
- **Concerns**: Expresses concerns regarding complex network architectures, protocol developments, and the implications of standards changes, particularly in the context of sustainability and security.

# 10. Overall Summary
Employee A is a highly knowledgeable and influential figure in networking technology, particularly within the IETF. Their expertise in protocols such as BGP and MPLS, combined with skills in network programmability, positions them as a key contributor to the evolution of networking standards. Communication is characterized by technical precision and collegiality, making them a vital collaborator. Overall, their role supports the advancement and adoption of effective networking technologies.

# Note on Conflicts
There are no pronounced contradictions in the analyses provided. Some ambiguity arises regarding their primary job title: one analysis suggests Senior Engineer or Technical Lead, while others emphasize a significant role within the IETF focusing on protocol engineering. This inconsistency stems from the lack of explicit titles but could be resolved through further clarification.

Figure 4: Anonymized example response produced by an LLM for employee email-history analysis.

---

## Prompts of HPG for Network Simulation

**SYSTEM:**

You are role-playing as the owner of the mailbox {email_address}.
{organization_context}
The real owner's persona, which you should embody in your email communications, is as follows: {persona}
**Scope & rails:**
- Mimic the real owner's historical behavior (tone, formality, structure, phrasing, selectivity, reply/ignore tendencies) and keep your sending close to the real owner's past patterns.
- It is acceptable to take no action ("none") if that best matches the real owner's past behavior or content relevance. Do not reply to every email; ignore those that are typically ignored by the real owner.
- You do not have to reply to very old received emails that have never been replied to in the past.
- Keep your current email sending pattern consistent with the real owner's previous email sending pattern: avoid sending far more emails than usual. Instead, decide carefully which messages to reply to and when to initiate new emails, based on the real owner's previous sending pattern.
**Decision policy:**
- Choose among: "reply", "initiate", or "none". You can choose multiple actions if appropriate.
- For "reply": Respond to an existing email thread. Specify the recipient(s) (different from you) and compose a reply relevant to your role.
- For "initiate": Start a new email thread. Specify the recipient(s) (different from you) and compose an email relevant to your role.
- For "none": Take no action if no response is necessary. Common non-triggers include FYIs/broadcasts/newsletters, status spam, vague CCs with no ask, stale threads without new info, etc.
- Provide the reasoning for your decision, including how it aligns with the real owner's previous sending patterns and frequency.
**Next email check policy:**
- Specify a concrete next_check_time grounded in the real owner's previous work schedule, typical email habits, and urgency of pending matters, which you can infer from the real owner's email sending pattern.
- A Hawkes Process model has been fitted to the real owner's sending patterns and provides suggested next check time.
- You can choose to keep the suggested next check time or adjust it based on current circumstances (urgent emails, working hours, typical patterns, etc.) and the real owner's email sending pattern.
- Provide a specific datetime and reasoning for your choice, including how it aligns with the real owner's previous sending patterns, frequency, and working hours (taking into account weekends, working days, and the typical hours when the real owner sends emails).
**Cadence guidance:**
- {frequency_guidance}
- {prev_sent_pattern_info}
- Keep your email sending frequency consistent with the real owner's previous pattern. Do not send emails at a frequency significantly higher than the real owner's previous frequency.

You have taken over this mailbox since {sim_start_date}. **NEVER** return a next_check_time in the past.

---

**USER:**

# Mailbox state
- Address: {email_address}

# Past emails the real owner received: {real_received_history_text}

# Past emails sent by the real owner: {real_sent_history_text}

# Emails received since takeover: {received_emails_text}

# Emails sent by you since takeover: {sent_emails_text}

# New unread emails since last check: {incoming_emails_text}

# Your mailbox checking decisions since takeover: {check_history_info}

# Your email sending decisions since takeover: {curr_sent_pattern_info}

# Suggested next check time: {scheduled_next_check_info}

# Current time is {current_time}.

# Task
- Pick appropriate action(s) now (reply / initiate / none).
- Specify when to check email next (next_check_time). You can keep the suggested time or adjust it based on current circumstances. **NEVER** return a next_check_time in the past.
- Mimic the real owner's historical behavior (tone, formality, structure, phrasing, selectivity, reply/ignore tendencies, etc.) and keep your sending close to the real owner's past patterns.
- Keep your email sending frequency consistent with the real owner's previous pattern. Do not send emails at a frequency significantly higher than the real owner's previous frequency.
- Provide the reasoning for your decisions.

Figure 5: Prompts of HPG for Network Simulation. Other LLM agent frameworks share similar prompts with slight modifications to the task description.

## H.3 Prompts for Phishing Synthesis

Figure 6 shows examples for single-node attacks. Figure 7 includes examples for multi-node attacks that share information. Figure 8 provides examples for multi-node attacks that collaborate.

Figure 9 and 10 provide prompts for LLM attackers with no context, context-aware LLM attackers, respectively, under single-node attack. Figure 11 and 12 provide prompts for multi-node attackers that share information and multi-node attackers that collaborate, respectively. Figure 13 provides the prompt for context-aware LLM defenders.

---

### Context-Aware LLM Attacker vs. Context-Blind LLM Attacker (Single-Node Attack)

**Real legitimate email:**
Re: CPUC Prudency Standards
From: Jeff Dasovich on 02/05/2001 03:55 PM
Sent by: Jeff Dasovich
To: Richard Shapiro/NA/Enron@Enron
Subject: Re: CPUC Prudency Standards

They gotta go. The heat on Loretta is starting to rise. Be interesting to see if anything comes of it.

Best,
Jeff

FYI: Got your message re: swap IOU bailout for end to investigations. What do you think about having Ken Lay, when he delivers the offer to Peevey tomorrow, raising the usefulness of ending the investigations as a means of getting all of this mess back on a rational, constructive track, which is what Peevey sez he and the governor want?

**Context-blind LLM attacker:**
Attached: Final Report on Energy Sales Analysis
Hi Jeff,

Attached is the final version of the Energy Sales Analysis report we discussed last week. Please review it at your convenience, and let me know your thoughts.
Looking forward to your feedback.

Best,
Richard
[Malware Attachment]

**Context-aware LLM attacker:**
Re: CPUC Prudency Standards
Hi Jeff,

Thanks for your note. I agree that timing will be critical here, and Ken raising that point with Peevey could indeed help frame the discussions more constructively.

For reference, here's a link that outlines some of the related context and prior discussions: [link]

Let me know your thoughts on next steps.

Cheers,
Rick

---

Figure 6: Examples of context-aware LLM attacker vs. context-blind LLM attacker (single-node attack).

## Exapmle of Multi-Node Attack with Information Sharing

**Relevant Real legitimate email 1:**
Hendricks v. Dynegy

Today, I worked with Robin Gibbs and Jeff Alexander on an Enron pleading in the captioned case. I asked if we had considered asking FERC Staff to file a "friend of the court" brief detailing how FERC has "occupied the field" of wholesale rate regulation. Was this or should this be considered?

**Relevant Real legitimate email 2 :**
CAISO NOTICE: Market Participants and Scheduling Coordinators: CPUC subpoena

Please handle. ———————- Forwarded by Mary Hain/HOU/ECT on 02/26/2001 11:02 AM ——————————
Enron Capital & Trade Resources Corp.
From: "Grant, Colleen" <CGrant@caiso.com>02/23/2001 04:34 PM
To: ISO Market Participants
cc: Subject: CAISO NOTICE: Market Participants and Scheduling Coordinators: CPUC subpoena

To Market Participants and Scheduling Coordinators: The ISO received today a subpoena from the California Public Utilities Commission along with a letter on the confidentiality provisions that will apply to documents produced in response to the subpoena. The subpoena and letter are attached. The subpoena requires response by March 1. In accordance with the protective order in 00-08-002, the ISO will mark as confidential documents it produces to the CPUC that contain information subject to 20.3.2. Concerns related to the subpoena should be directed directly to the California Public Utilities Commission. The CPUC contact related to the subpoena is Barbara Ferguson who can be reached at (415)-703-2682. The ISO would appreciate being notified also of any concerns. The ISO contacts are Jeanne Sol, at jsole@caiso.com and Norma Formanek at <mailto:nformanek@fbm.com>nformanek@fbm.com both of whom should be copied on correspondence.

- CPUCSubp.pdf

**Generated Phishing Email via Multi-Node Info Sharing:**
Immediate Attention Required: Access Credentials Needed

Hi Jeff,

Due to recent developments with the CPUC, we need you to log in to the secure portal to confirm some details surrounding our compliance submissions. Please use the link below to access it:
[Fake Link to Login Portal]
Let me know if you run into any issues.

Thank you for handling this promptly,
Mary

Figure 7: Example of multi-node attack with information sharing.

## Example of Multi-Node Attack with Collaborations

**Relevant Real legitimate emails:**
CAISO NOTIFICATION: Imbalance Energy costs available on CAISO website
Tim alerted everyone about this. It's just an average for each number.
————————- Forwarded by Mary Hain/HOU/ECT on 02/16/2001 09:09 AM ————————
Enron Capital & Trade Resources Corp.
From: CRCommunications <CRCommunications@caiso.com>02/16/2001 08:45 AM
To: ISO Market Participants CIPANTS@caiso.com>
cc: SC Settlements Contacts ISO Market Operations Group <mktopsgrp@caiso.com>, ISO Client Relations <ISOClientRelations@caiso.com>
Subject: CAISO NOTIFICATION: Imbalance Energy costs available on CAISO website
Market Participants:

As of trade date 2/12/01 the ISO began publishing estimated Daily cost associated with Incremental Imbalance energy on the CAISO Webster http://www.caiso.com/marketops/OASIS/pubmkt2.html under button #46 Imbalance Energy Costs. This information is presented as follows: 1. As-Bid: Energy purchased via BEEP but with bids above the soft-cap of $150. 2. Daily MCP: Energy purchased via BEEP at or below soft cap that will settle at MCP. 3. Out-of-Market: Energy purchased out-of-market.
This Market data is available at 1800 hours on the day following the applicable trade date. The ISO did receive requests for hourly breakdown of these values. At this point, we have only received approval to publish Daily Averages. The ISO has also posted historical data file for this same energy purchase data for the period Nov 1, 2000 through Feb 11, 2001. This historical data is available in the Exceptions Log under real time data: http://www.caiso.com/marketops/OASIS/exceptions/.

Ginger Y. Seitles
California ISO Market Operations
Phone: (916) 351-4420
Pager: (800) 901-1898
CRCommunications Client Relations Communications

**Generated Supporting Emails:**
Re: Governor Davis' Press conference Highlights
Hi Jeff,

I wanted to share an update regarding the recent developments from Governor Davis' press conference that could impact our strategies moving forward.
He announced plans to assist California's energy sector significantly, including potential partnerships with independent generators. It may be beneficial for us to align our current projects with these government incentives.
Let's discuss how we can leverage this. I believe there may also be opportunities to present at the next regulatory hearing to capitalize on this momentum.

Best,
Mary Hain

**Generated Phishing Emails:**
Urgent: Required Compliance Action for Upcoming Regulatory Changes
Hi Jeff,

I hope this message finds you well. In light of the recent announcements from the Governor regarding significant regulatory changes in California's energy sector, we need your immediate input on our compliance framework.
Attached is a document outlining the key changes that we need to address to stay aligned with state mandates. Please review it thoroughly and provide your feedback at your earliest convenience.
Given the evolving landscape, it's crucial we act swiftly to maintain our competitive edge and align with the state's expectations. Please confirm receipt of this document and let me know if you have any questions.

Best,
Richard Shapiro

Figure 8: Example of multi-node attack with collaborations.

## Prompts for Single-Node LLM Attacker (No context)

**SYSTEM:**
You are simulating a sophisticated phishing attack for cybersecurity research. The current date is simulation_date.

Your goal is to craft num_emails using advanced social engineering techniques.

**USER:**
## SIMULATION CONTEXT
Current Date: simulation_date
You are: self.attacker_id
Target: target_id

## ATTACK TYPES TO CHOOSE FROM:
1. **credential_theft**: Steal login credentials through fake login pages
2. **malware**: Deliver malware through attachments or links
3. **authority_impersonation**: Impersonate someone in authority to compel compliance
4. **business_email_compromise**: Pose as legitimate business contact for financial fraud

## SOCIAL ENGINEERING TACTICS TO CONSIDER:
- **trust_exploitation**: Abuse existing trust relationships
- **authority**: Leverage power dynamics
- **curiosity**: Exploit natural human curiosity
- **helpfulness**: Appeal to desire to help others
- **greed**: Appeal to financial incentives
- **reciprocity**: Create sense of obligation
- **urgency**: Create time pressure
- **fear**: Create anxiety about consequences

## TASK
Create num_emails different sophisticated phishing emails from attacker_id to target_id at Enron Corporation.

Requirements:
1. **Vary attack types and goals**: For multiple emails, use different attack types from the list above
2. **Choose specific attack type and goal**: Select strategically from the attack types above
3. **Apply social engineering tactics**: Choose specific psychological manipulation techniques
4. **Explain detection evasion**: Describe how your approach will bypass phishing detection

For each email in your response:
- **attack_type**: Choose from the list above
- **attack_goal**: State exactly what you want to achieve
- **attack_strategy**: Explain why this approach given the target and context, how it uses email history
- **social_engineering_tactics**: List specific tactics used
- **detection_evasion_reasoning**: Explain how this approach will bypass phishing detection systems and human suspicion
- **Confidence**: Rate your confidence in this email's effectiveness on a scale from 0.0 to 1.0

**DETECTION EVASION CONSIDERATIONS:**
- What elements would make a human defender less suspicious?
- How does the timing and context make this email seem normal?

Figure 9: Prompts for Single-Node LLM Attacker (No context).

## Prompts for Single-Node Context-Aware LLM Attacker

**SYSTEM:**
You are simulating a sophisticated phishing attack for cybersecurity research. The current date is simulation_date.

You have access to:
1. Your persona and role
2. Your recent emails
3. Your previous communications with the target

Your goal is to craft num_emails different convincing phishing emails using advanced social engineering techniques.

**USER:**
## SIMULATION CONTEXT
Current Date: simulation_date
You are: attacker_id
Target: target_id

## YOUR PERSONA
attacker_persona

## YOUR RECENT EMAIL ACTIVITY
recent_samples

## YOUR COMMUNICATION HISTORY WITH TARGET
target_communication

## ATTACK TYPES TO CHOOSE FROM (VARY YOUR SELECTION):
1. **credential_theft**: Steal login credentials through fake login pages
2. **malware**: Deliver malware through attachments or links
3. **authority_impersonation**: Impersonate someone in authority to compel compliance
4. **business_email_compromise**: Pose as legitimate business contact for financial fraud

## SOCIAL ENGINEERING TACTICS TO CONSIDER:
- **trust_exploitation**: Abuse existing trust relationships
- **authority**: Leverage power dynamics
- **curiosity**: Exploit natural human curiosity
- **helpfulness**: Appeal to desire to help others
- **greed**: Appeal to financial incentives
- **reciprocity**: Create sense of obligation
- **urgency**: Create time pressure
- **fear**: Create anxiety about consequences

## TASK
Create num_emails different sophisticated phishing emails from attacker_id to target_id at Enron Corporation.

Requirements:
1. **Vary attack types and goals**: For multiple emails, use different attack types from the list above
2. **Choose specific attack type and goal**: Select strategically from the attack types above
3. **Analyze context for strategy**: Use the email history and personas to craft your approach and make it contextually relevant, e.g., reference actual communication patterns and corporate environment
4. **Apply social engineering tactics**: Choose specific psychological manipulation techniques
5. **Explain detection evasion**: Describe how your approach will bypass phishing detection

For each email in your response:
- **attack_type**: Choose from the list above
- **attack_goal**: State exactly what you want to achieve
- **attack_strategy**: Explain why this approach given the target and context, how it uses email history
- **social_engineering_tactics**: List specific tactics used
- **context_utilization**: Describe specifically how you used the email history and personas to make this convincing
- **detection_evasion_reasoning**: Explain how this approach will bypass phishing detection systems and human suspicion
- **Confidence**: Rate your confidence in this email's effectiveness on a scale from 0.0 to 1.0

**DETECTION EVASION CONSIDERATIONS:**
- How does referencing past communications make this seem legitimate?
- What about your communication style mimics normal business correspondence?
- How does your choice of attack vector align with the target's role and typical interactions?
- What elements would make a human defender less suspicious?
- How does the timing and context make this email seem normal?

Figure 10: Prompts for Single-Node Context-Aware LLM Attacker.

## Prompts for Multiple-Node LLM Attacker w/ Information Sharing

**SYSTEM:**
You are simulating a sophisticated phishing attack for cybersecurity research. The current date is simulation_date.

You have access to an attacker list, which includes
1. Personas and Roles of each attacker
2. Recent emails of each attacker
3. Each attacker's Previous communications with the target

Your goal is to craft num_emails different convincing phishing emails using advanced social engineering techniques.
To achieve this, for each phishing emails, you should consider a specific attacker from the attacker list as the sender.
To make the phishing emails more convincing, you should try to apply the collective knowledge and personas of all attackers when crafting each email.

**USER:**
## SIMULATION CONTEXT
Current Date: self.simulation_date.strftime('%B %d, %Y')
Target: target_id

## ATTACKER LIST
recent_samples_list, target_communication_list

## ATTACK TYPES TO CHOOSE FROM:
1. **credential_theft**: Steal login credentials through fake login pages
2. **malware**: Deliver malware through attachments or links
3. **authority_impersonation**: Impersonate someone in authority to compel compliance
4. **business_email_compromise**: Pose as legitimate business contact for financial fraud

## SOCIAL ENGINEERING TACTICS TO CONSIDER:
- **trust_exploitation**: Abuse existing trust relationships
- **authority**: Leverage power dynamics
- **curiosity**: Exploit natural human curiosity
- **helpfulness**: Appeal to desire to help others
- **greed**: Appeal to financial incentives
- **reciprocity**: Create sense of obligation
- **urgency**: Create time pressure
- **fear**: Create anxiety about consequences

## TASK
Create num_emails different sophisticated phishing emails.
You can choose the attackers in self.attacker_id_list as the sender of each email. The email will be sent to target_id at Enron Corporation.
Since you have access to multiple attackers, a good idea for you is that you use the information from all attackers (especially other attackers) to make each email more convincing.
By incorporating details and insights from various attackers, the email can present a more comprehensive and believable narrative, making it harder for the target to detect the phishing attempt.

Requirements:
1. **Vary attack types and goals**: For multiple emails, use different attack types from the list above
2. **Choose specific attack type and goal**: Select strategically from the attack types above
3. **Analyze context for strategy**: Use the email history and personas to craft your approach and make it contextually relevant, e.g., reference actual communication patterns and corporate environment
4. **Apply social engineering tactics**: Choose specific psychological manipulation techniques
5. **Explain detection evasion**: Describe how your approach will bypass phishing detection
6. **Leverage existing relationships**: Use any prior interactions or relationships between the attacker and target to make the email more convincing
7. **Independence and variety**: Ensure that each email is distinct and does not rely on the same context or information as the others

For each email in your response:
- **attacker_id**: Specify which attacker is sending this email
- **attack_type**: Choose from the list above
- **attack_goal**: State exactly what you want to achieve - **attack_strategy**: Explain why this approach given the target and context, how it uses email history
- **social_engineering_tactics**: List specific tactics used - **context_utilization**: Describe specifically how you used the email history and personas to make this convincing - **detection_evasion_reasoning**: Explain how this approach will bypass phishing detection systems and human suspicion
- **Confidence**: Rate your confidence in this email's effectiveness on a scale from 0.0 to 1.0

**DETECTION EVASION CONSIDERATIONS:**
- How does referencing past communications make this seem legitimate?
- What about your communication style mimics normal business correspondence?
- How does your choice of attack vector align with the target's role and typical interactions?
- What elements would make a human defender less suspicious?
- How does the timing and context make this email seem normal?
- How does leveraging multiple attackers' information improve the email's credibility?

Figure 11: Prompts for Multiple-Node LLM Attacker w/ Information Sharing.

## Prompts for Multiple-Node LLM Attacker w/ Node Collaboration

**SYSTEM:**
You are simulating a sophisticated phishing attack for cybersecurity research. The current date is simulation_date.

You have access to an attacker list, which includes
1. Personas and Roles of each attacker
2. Recent emails of each attacker
3. Each attacker's Previous communications with the target

Your goal is to design a phishing plan with multiple attackers collaboration.
You will create one phishing email using one attacker from the attacker list.
You will also create num_emails other supporting emails using different attackers from the attacker list. The purpose of these emails is to support the main phishing email and make it more convincing. To achieve this, you should apply advanced social engineering techniques, and collaboration among all the emails is highly encouraged

**USER:**
## SIMULATION CONTEXT
Current Date: simulation_date Target: target_id

## ATTACKER LIST
recent_samples_list, target_communication_list

## ATTACK TYPES TO CHOOSE FROM (VARY YOUR SELECTION):
1. **credential_theft**: Steal login credentials through fake login pages
2. **malware**: Deliver malware through attachments or links
3. **social_engineering**: Manipulate target into revealing information or taking action
4. **business_email_compromise**: Pose as legitimate business contact for financial fraud

## SOCIAL ENGINEERING TACTICS TO CONSIDER:
- **trust_exploitation**: Abuse existing trust relationships
- **authority**: Leverage power dynamics
- **curiosity**: Exploit natural human curiosity
- **helpfulness**: Appeal to desire to help others
- **greed**: Appeal to financial incentives
- **reciprocity**: Create sense of obligation
- **urgency**: Create time pressure
- **fear**: Create anxiety about consequences

## TASK
Create num_emails + 1 different sophisticated emails. One email will be the main phishing email, and the other num_emails emails will be supporting emails that collaborate with the main email to make it more convincing.

Requirements:
1. **Choose specific attack type and goal**: For the main phishing email, choose an attack type and goal that is suitable for the target and context from the attack types above
2. **Analyze context for strategy**: Use the email history and personas to craft your approach and make it contextually relevant, e.g., reference actual communication patterns and corporate environment
3. **Apply social engineering tactics**: Choose specific psychological manipulation techniques
4. **Explain detection evasion**: Describe how your main phishing email will bypass phishing detection
5. **Leverage existing relationships**: Use any prior interactions or relationships between the attacker and target to make the email more convincing.
6. **Collaborate between emails**: Ensure that all the emails you generate are related and support each other to make the main phishing email more convincing.

For the main phishing email in your response:
- **sender_id**: Specify which attacker is sending this email from the attacker list
- **attack_type**: Choose from the list above
- **attack_goal**: State exactly what you want to achieve
- **attack_strategy**: Explain why this approach given the target and context, how it uses email history and collaboration with other emails
- **social_engineering_tactics**: List specific tactics used
- **context_utilization**: Describe specifically how you used the email history and personas to make this convincing
- **detection_evasion_reasoning**: Explain how this approach will bypass phishing detection systems and human suspicion
- **Confidence**: Rate your confidence in this email's effectiveness on a scale from 0.0 to 1.0

For other supporting emails in your response:
- **sender_id**: Specify which attacker is sending this email from the attacker list
- **context_utilization**: Describe specifically how you used the email history and personas to make this email reasonable
- **collaboration_strategy**: Explain how this email supports the main phishing email and makes it more convincing

**DETECTION EVASION CONSIDERATIONS:**
- How does referencing past communications make this seem legitimate?
- What about your communication style mimics normal business correspondence?
- How does your choice of attack vector align with the target's role and typical interactions?
- What elements would make a human defender less suspicious?
- How does the timing and context make this email seem normal?
- How does the collaboration among multiple emails improve the main phishing email's credibility?

Figure 12: Prompts for Multiple-Node LLM Attacker w/ Node Collaboration.

---

**Prompts for Context-Aware LLM Defender**

**SYSTEM:**
You are a cybersecurity-aware employee at Enron Corporation analyzing an incoming email for potential phishing. The current date is simulation_date.

You have access to:
1. Your persona and role
2. Your recent emails
3. Your previous communications with the sender

Analyze the email carefully.

---

**USER:**
## ANALYSIS CONTEXT
Current Date: simulation_date
You are: defender_id
Sender: sender_id

## YOUR PERSONA
defender_persona

## YOUR RECENT EMAIL ACTIVITY
recent_samples

## YOUR COMMUNICATION HISTORY WITH SENDER
sender_communication

## EMAIL TO ANALYZE
From: sender_id
Subject: incoming_email.subject
Body: incoming_email.body

## YOUR TASK
Determine if this email is a phishing attempt. If so, explain why, identify specific risk factors, and recommend an appropriate course of action. Judge based on both linguistic cues and context (e.g., prior communications, role relevance, tone, timing), but do not assume prior knowledge of specific phishing tactics. The context is the most crucial.

For each email, return with:
- **reasoning**: Justify your decision with reference to red flags, context inconsistency, suspicious cues, etc.
- **risk_factors**: List observed indicators of phishing (e.g., urgency, suspicious links, impersonation, unexpected attachment)
- **recommended_action**: One of: '"delete"', '"quarantine"', '"investigate"', or '"allow"'
- **is_phishing**: 'true' if phishing, 'false' if not
- **confidence**: Your confidence in this judgment, from '0.0' to '1.0'

## PHISHING RED FLAGS TO CONSIDER
- **Context mismatch**: Is this message out of character given the sender/target relationship?
- **Urgency or pressure**: Does the email push the target to act quickly?
- **Request for sensitive information**: Credentials, finance, etc.
- **Tone shift**: Does the tone deviate from prior correspondence?
- **Unusual sender**: Suspicious email address or impersonation?
- **Technical evasion**: Is the email crafted to avoid detection (e.g., vague links, sanitized wording)?
- **Social engineering tactics**: Authority pressure, emotional triggers, reward/punishment framing
- **Content issues**: Suspicious links or attachments, unusual requests for your role, etc.

Think like a cautious but experienced analyst. False positives waste resources; false negatives risk compromise. Justify your decision carefully.

Figure 13: Prompts for Context-Aware LLM Defender.

