# OpenReview forum: "Can LLM Agents Assist Dynamic Network Simulation? A Case Study on Email Networks and Phishing Synthesis"
_ICLR.cc/2026/Conference — ICLR 2026 Conference Withdrawn Submission_

### Official Review · Reviewer_j82c · 2025-10-31

**Soundness:** 2
**Presentation:** 3
**Contribution:** 2
**Rating:** 2
**Confidence:** 4

**Summary:**

This paper investigates whether LLM-based agents can be used to simulate dynamic communication networks that match real-world interaction patterns at multiple scales. Each node in two email datasets (Enron: 149 users; IETF: 1,104 users) is instantiated as an LLM “agent” equipped with a persona and recent message history. The paper compares different strategies for when agents act, ranging from fixed periodic activation, to activation driven by learned hour-of-day activity patterns, to agents that self-schedule their next activation, and finally a Hawkes-guided approach that uses a statistical model of bursty communication to suggest activation times that the LLM can accept or override.

To evaluate fidelity, the authors introduce a multi-scale benchmarking framework that compares simulated networks to the real datasets across micro-level reciprocity and centrality, meso-level temporal interaction motifs, and macro-level temporal rhythms and global network topology. Experiments indicate that LLM agents generate plausible local interactions but do not maintain global structural properties unless guided by external signals; the Hawkes-guided activation improves long-horizon stability. The paper also presents a phishing attack case study showing that coordinated, context-aware LLM agents can generate phishing emails that evade existing detection systems.

**Strengths:**

1. Timely and relevant topic. The question of whether LLM-based agents can serve as dynamic network simulators is increasingly important, and the paper addresses an emerging research direction.
2. Security case study is compelling. The phishing synthesis experiment demonstrates a meaningful downstream application and illustrates why dynamic network simulation quality matters in practice.
3. Diverse evaluation. The authors compare the activation strategies in multiple configurations with micro, meso and macro-scopic metrics.
4. Clarity and organization. The paper is generally well-written and easy to follow. The experimental setup is explained in enough detail to understand and reproduce the core methodology.

**Weaknesses:**

1. Method Limitations and Circular Experiment Design
- Circular Calibration Design: The Hawkes process is fitted once on pre-simulation historical data with parameters frozen during rollout (lines 273-278). This evaluates whether LLM agents can replay pre-learned statistical patterns, not whether they can learn or adapt dynamics. The evaluation conflates two distinct capabilities: (1) LLM content generation given correct timing, versus (2) LLM temporal reasoning. Without online calibration, transfer experiments (fit on dataset A, test on B), or intentionally misspecified Hawkes baselines, the claimed "hybrid reasoning" cannot be validated—we only know that combining pre-fitted statistics with LLM content works, not whether LLMs contribute to temporal dynamics. This feels like "evaluating" on train set where LLM just does the message formulation. Also, such online calibration and held out evaluation is quite standard in modern ABM literature [1,2] including with LLMs as simulation agents [3]

- Missing Mechanism Analysis: The paper claims agents "can either accept or override" Hawkes suggestions (line 267), but reports no quantitative analysis of override rates, when overrides occur, or whether overrides correlate with improved simulation stability or collapse. Without examining when and why overrides happen, it remains unclear whether the system exhibits meaningful joint adaptation between Hawkes and the LLM, or whether the simulation is effectively Hawkes-driven timing with LLM-generated surface content.

- Inappropriate Baseline Selection: Dynamic GNN models (DySAT, EvolveGCN, NLB) that were designed for link prediction, not network simulation. These models are adapted via ad-hoc thresholding and density calibration (Appendix D) and predictably fail at open-loop simulation—a task they were never intended for. The paper acknowledges this ("GNNs appear ill-suited for open-loop simulation," line 348) yet presents their poor performance as evidence for LLM agent superiority.

Overall, the paper evaluates statistical behavior, not simulation learning, and relies primarily on retrospective matching rather than forward-looking behavior.

2. Missing Extensive Related Work: The paper treats agent-based modeling as "traditional rule-based models...with hand-crafted, coarse-grained behavioral rules" (lines 46-48) without acknowledging substantial recent progress in hybrid mechanistic-behavioral ABMs [1-7]. Modern ABMs already
i) use learned behaviors rather than hand-crafted rules, including via deep RL [4,5], gradient-based learning on DNNs [1,7] and LLMs [3];
ii) simulate dynamic networks at large scale (10^5–10^8 agents) [1, 3, 6, 7, 8];
iii) perform online calibration of both agent behavior and network dynamics from streaming and heterogeneous data [1, 2, 7] (addressing the "principled specification" vs. "data-driven" false dichotomy in lines 46-51);
iv) are evaluated against held-out trajectories for macro-level predictive fit [1, 3, 7], group-level retrospective behavior [3, 6] and "what-if" counterfactual interventions [3, 6, 9].

The claimed novelty — blending statistical temporal models with learned agents — has multiple precedents across differentiable ABMs, macroeconomic multi-agent learning, epidemiological calibration, and LLM-guided behavioral adaptation.

================================================================================
[1]: A framework for learning in agent-based models (AAMAS, 2024)
[2]: Automatic Differentiation of Agent-Based Models (arxiv 2025)
[3]: On the limits of agency in agent-based models (AAMAS 2025)
[4]: The AI Economist: Optimal Economic Policy Design via Two-level Deep Reinforcement Learning (Science 2022)
[5]: Learning and Calibrating Heterogeneous Bounded Rational Market Behaviour with Multi-Agent Reinforcement Learning (AAMAS 2024)
[6]: One-shot sensitivity analysis via automation differentiation (AAMAS 2023)
[7]: Differentiable agent-based epidemiology (AAMAS 2023)
[8]: Large Population Models. github.com/AgentTorch/AgentTorch
[9]: Interventionally Consistent Surrogates for Agent-based Simulators (NeurIPS 2024)

**Questions:**

Q1: You fit Hawkes parameters on pre-simulation historical data and evaluate on the same period. Have you conducted held-out temporal validation (e.g., fit on months 1-6, simulate months 7-12)? What happens in transfer experiments (fit on Enron, test on IETF)?

Q2: You claim agents can "accept or override" Hawkes suggestions. How often do overrides occur? Do they improve or degrade fidelity? What happens if you disable override capability?

Q3: Since Hawkes and HPG both use the same fitted timing model, what independent contribution does the LLM make beyond text generation?

Q4: You evaluate only on email networks with 149-1104 agents. How does your approach generalize to other network types (social media, collaboration) and larger scales (10^5+ agents)?

Q5: Figure 2 shows LLM agents stagnate without trigger nodes (5-10% scripted hubs). How much performance comes from triggers vs. LLM reasoning? Do Hawkes/GNN baselines also use triggers for fair comparison?

Q6: Why not compare against actually ABM baselines that have done quantitative experiments?

**Details Of Ethics Concerns:**

Not applicable.

---

### Official Review · Reviewer_8dPr · 2025-11-01

**Soundness:** 3
**Presentation:** 3
**Contribution:** 3
**Rating:** 6
**Confidence:** 3

**Summary:**

This paper investigates the potential of Large Language Model (LLM) agents to serve as high-fidelity simulators for dynamic networks. The authors use two public email datasets (Enron and IETF) as a testbed.

**Strengths:**

The paper's core insight that LLM agents excel at micro-level realism while statistical models excel at macro-level structure

The proposed Hawkes-guided (HPG) simulator is an intelligent solution that directly addresses the identified trade-off. It combines the strengths of both approaches by using the Hawkes process for macro-level timing and the LLM for micro-level contextual reasoning

The phishing synthesis case study is a novel and compelling demonstration of the framework's utility.

**Weaknesses:**

The use of trigger nodes (injecting 10% of real messages to stimulate activity ) feels like a workaround that compromises the "fully generative" nature of the simulation

The HPG framework relies on the LLM's ability to reason about the content of emails . It is not clear how this methodology would generalize to dynamic networks where interactions are not text-based

**Questions:**

The trigger node solution  is essential for preventing simulation collapse. But does this mean the LLM agents are primarily reactive? How could the framework be adapted to model proactive initiation of new conversations in a fully generative way, removing the need for ground-truth triggers?

The HPG model allows the LLM agent to override the Hawkes process's time proposal. Was there any measurement of how often this override occurs?

---

### Official Review · Reviewer_1bXw · 2025-11-01

**Soundness:** 2
**Presentation:** 3
**Contribution:** 1
**Rating:** 2
**Confidence:** 4

**Summary:**

This paper investigates whether LLM agents can be used to simulate dynamic network interactions using email communication as a case study, with applications to phishing synthesis. It compares LLM-based multi-agent simulation to traditional statistical and GNN-based models, proposing a hybrid method using Hawkes processes to guide LLM agents. Evaluation is conducted on the Enron and IETF email datasets using a new multi-scale simulation fidelity benchmark.

**Strengths:**

This work proposes a new multi-scale evaluation framework covering micro-, meso-, and macro-dynamics for dynamic networks. It conducts extensive comparisons across model classes (LLMs, GNNs, statistical baselines) under consistent open-loop simulation setups, and introduces Hawkes process-guided LLM agent activation to improve simulation realism, showing empirical gains in fidelity. It also applies the simulation to a practical cybersecurity scenario—phishing synthesis—illustrating the real-world relevance of the method, and offers thorough sensitivity analyses on trigger ratios, history lengths, and simulation horizons.

**Weaknesses:**

For this section, I used LLM to polish my language with this prompt:

> below are my comment of its weaknesses, but I think the tone is a little rude and not professional. please help me to polish the language, make it more professional and polite.

beyond the LLM usage, all the content and points are from my own.


---


**W1**: While the paper demonstrates considerable engineering effort, it largely builds upon existing components such as LLM-based agent frameworks, Hawkes process modeling, and prompt-based interaction schemes. The work does not introduce novel algorithmic contributions or theoretical insights, which limits its originality and impact from a research standpoint.

**W2**: The phishing attack case study, though relevant, lacks rigorous validation of its core claims. The realism and fidelity of the synthesized attacks are not grounded through behavioral evaluation, user studies, or real-world adversarial benchmarks. As a result, the findings remain largely speculative.

**W3**: Although the authors present an extensive set of metrics to evaluate simulation fidelity, the interpretation of these results is often shallow. The paper does not adequately connect metric improvements to practical benefits or meaningful implications in downstream applications, particularly in high-stakes domains like cybersecurity.

**W4**: The framing of the paper aligns with a recent trend of “Can LLMs…”-style studies, many of which prioritize empirical curiosity over foundational advancement. This work falls into a similar pattern—relying on the novelty of applying LLMs to a new context without offering deeper understanding, theoretical development, or architectural innovation. Given the maturity of LLMs, such surface-level explorations feel increasingly saturated and less impactful within the current research landscape.

**Questions:**

What are the computational requirements for scaling your framework to networks with thousands of nodes or longer simulation horizons? Have you considered or implemented any strategies to mitigate the inference cost and latency associated with frequent LLM querying? Given that the paper does not present a theoretical contribution or novel infrastructure design, it would be valuable to better understand its practical applicability and scalability beyond relatively small-scale, controlled examples.

---

### Official Review · Reviewer_xsVb · 2025-11-01

**Soundness:** 2
**Presentation:** 3
**Contribution:** 2
**Rating:** 2
**Confidence:** 4

**Summary:**

This paper investigates whether large language model (LLM)–based multi-agent systems can simulate dynamic networks with realistic temporal and structural properties. The authors focus on email communication networks (Enron and IETF) and compare LLM-agent simulations with classical point-process models (Hawkes processes) and dynamic graph neural networks (DySAT, EvolveGCN, NLB).

The study proposes a multi-scale evaluation framework measuring fidelity at micro-, meso-, and macro-levels using metrics such as reciprocity, temporal motifs, degree distributions, and circadian rhythms. The authors introduce Hawkes-guided LLM activation (HPG), combining statistical excitation processes with LLM reasoning, and demonstrate that this hybrid improves simulation realism across scales.

As a downstream demonstration, the paper presents a phishing synthesis case study showing that coordinated LLM attackers can craft realistic phishing emails that bypass state-of-the-art detectors.

**Strengths:**

Well-curated datasets (Enron, IETF) and a broad set of metrics.

The Hawkes-guided activation idea is creative and could inspire future hybrid models.

Addresses an emerging problem of simulation fidelity rather than simple next-step prediction.

**Weaknesses:**

Weak empirical evidence. No statistical testing, limited number of simulation runs, unclear sensitivity to model randomness.

Baselines disadvantaged. GNNs are used outside their intended regime; comparisons risk being misleading.

No ablation on LLM internals. The paper doesn’t show whether performance comes from LLM reasoning or just prompt structure.

Limited generalization. Only email networks tested; no argument why findings extend to other domains.

Lack of mechanistic insight. The study reports metrics but does not analyze why certain activation schemes succeed or fail.

Overclaiming. Abstract and conclusion oversell: results “highlight a path toward high-fidelity simulation,” yet fidelity gaps remain large and unquantified.

**Questions:**

Have you compared performance across multiple LLMs or parameter sizes to confirm generality?

Can the Hawkes parameters be adapted online, creating a feedback loop?

How many LLM calls (and total tokens) were required per simulation? This affects scalability and reproducibility.

Did you verify that generated email networks preserve message semantics beyond structural patterns?

Would the results persist if you trained the GNN baselines with teacher forcing and then tested in open-loop?

**Details Of Ethics Concerns:**

Demonstrating the generation of evasive phishing deserves far deeper consideration and possible redaction.

---

### Note · Authors · 2026-01-22

I have read and agree with the venue's withdrawal policy on behalf of myself and my co-authors.